# Study of Key Parameters and Uncertainties Based on Integrated Energy Systems Coupled with Renewable Energy Sources

Xin Liu [1,2], Yuzhang Ji [1,2], Ziyang Guo [1,2], Shufu Yuan [1], Yongxu Chen [1] and Weijun Zhang [1,2,*]

1   School of Metallurgy, Northeastern University, Shenyang 110819, China; 1810574@stu.neu.edu.cn (X.L.)
2   State Environmental Protection Key Laboratory of Eco-Industry, Northeastern University, Shenyang 110819, China
*   Correspondence: zhangwj@smm.neu.edu.cn

**Abstract:** The extensive research and application of integrated energy systems (IES) coupled with renewable energy sources have played a pivotal role in alleviating the problems of fossil energy shortage and promoting sustainability to a certain extent. However, the uncertainty of photovoltaic (PV) and wind power in IES increases the difficulty of maintaining stable system operation, posing a challenge to long-term sustainability. In addition, the capacity configuration of each device in IES and the operation strategy under different conditions will also significantly impact the operation cost and expected results of the system, influencing its overall sustainability. To address the above problems, this paper establishes an optimization model based on linear programming to optimize the equipment capacity and operation strategy of IES coupled with PV and wind power with the minimum total annual cost as the objective function, thereby promoting economic sustainability. Moreover, an integrated assessment framework, including economic, energy efficiency, and environmental aspects, is constructed to provide a comprehensive assessment of the operation of IES, ensuring a holistic view of sustainability. Finally, taking the IES of an industrial park in Xi'an, China, as the specific case, sensitivity analysis is used to explore the impact of a variety of critical parameters on the equipment capacity and operating strategy. Additionally, the Monte Carlo method is used to explore the impact of source-load uncertainty on the performance of the IES. The results show that the facilitating or constraining relationship between renewable energy access and the cascading utilization of combined heat and power generation (CHP) energy depends on the relative magnitude of the user load thermoelectric ratio to the prime mover thermoelectric ratio. To cope with the negative impact of source-load uncertainty on the stable operation of the IES, the capacities of the electric chiller and absorption chiller should be increased by 4.0% and 5.8%, respectively. It is worth noting that the increase in the penetration rate of renewable energy has not changed the system's dependence on the grid.

**Keywords:** integrated energy systems; sustainability; sensitivity analysis; optimize configuration

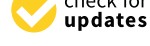



## 1. Introduction

With the continuous progress of industrialization, the shortage of fossil energy and environmental pollution have become global problems. In order to realize sustainable development, the global energy consumption structure has been gradually transformed in recent years, and the consumption of renewable energy in particular has grown significantly. Solar and wind power are the main contributors to new renewable energy capacity globally. According to the data released by the Statistical Yearbook of World Energy, the total global installed capacity of photovoltaic (PV) and wind power grew by a record 266 GW in 2022 [1]. Among them, China's PV and wind power capacity grew the most, accounting for 37% and 41% of global capacity additions, respectively. By the end of April 2023, China's total installed capacity of PV and wind power reached 820 GW, accounting for 31% of the country's total installed power generation capacity, and it is expected that the total

installed capacity will reach more than 1200 GW by 2030 [2]. Although PV and wind power have grown significantly in terms of installed capacity, in actual operation, PV and wind power are largely affected by environmental changes, resulting in greater uncertainty in output, which increases the challenge of their utilization [3]. Therefore, how to rationally and efficiently consume renewable energy represented by PV and wind power has become a hot topic at present.

In this context, integrated energy systems (IES) has become one of the new generations of energy systems that countries around the world are focusing on researching and developing because it has the advantage of realizing the complementary advantages of different kinds of energy sources, including renewable energy sources and fossil energy sources, as well as the advantage of the gradient utilization of energy, allowing it to effectively improve the efficiency of the overall use of energy [4]. In IES, there are multiple energy media, such as cold, heat, electricity, and gas, realizing complementarity and synergy between different energy media [5]. A variety of energy media integrated by the system, through optimized scheduling and synergistic management, can satisfy a variety of energy loads, such as cold, heat, electricity, and gas, for users. Generally, IES mainly includes units such as energy supply networks (e.g., cooling, heating, electricity, and gas pipeline networks), energy conversion devices (e.g., prime movers, chillers, PV, and wind power systems), energy storage units (e.g., heat storage tanks and storage batteries), user loads, etc. [6]. When coupling PV and wind power into IES, on the one hand, its own regulation capability can reduce the uncertainty of intermittent power generation caused by geographical and environmental constraints of PV and wind power, and improve the rate of consumption of renewable energy sources [7,8]. On the other hand, the access to PV and wind power can reduce the demand for fossil energy in IES, thus achieving the purpose of energy saving and environmental protection [9].

Although IES is an efficient and clean method of utilizing energy, the increases in complexity of equipment selection within the system, penetration rate of renewable energy on the power generation side and diversity of load demand on the customer side, the resulting diversity of equipment configuration and operation strategy selection, as well as source-load uncertainty and other issues, have created higher requirements for the safe, stable, and efficient operation of IES. In terms of these problems, the current research on the optimal design of IES generally focuses on several aspects, such as equipment capacity design, operation strategy optimization, source-load uncertainty study, and economic benefit evaluation.

Generally speaking, IESs are usually built according to regional characteristics, so it is necessary to first consider the local energy resources available, select relevant equipment according to the available energy, and carry out a reasonable design and selection of its capacity and type [10,11]. Sanaye et al. [12] used the maximum rectangle method to design an approximation of the total nominal power required by the gas prime mover in a combined cooling, heating, and electricity system for different scenarios. The same methodology was used to determine the capacity of the system equipment that should be used for residential use by Wang et al. [13]. Using ships as an application scenario, Cao et al. [14–16] proposed a mathematical planning-based approach to design the optimal configuration and scheduling strategy of the system, taking into account the effects of different user loads and weather conditions. In addition, there are usually a variety of user load demand, energy production, and conversion equipment in IES, so the study of the relationship between the user's various load demand and equipment operation strategy should not be ignored [17]. According to more than 200 application cases of IES investigated and studied by authors, it was found that some systems did not operate efficiently due to the lack of a reasonable operation program designed in the early stage, and some systems were even shut down due to low returns [18,19]. To achieve multi-energy, complementary, comprehensive energy systems between the various units, a reasonable strategy of operation that maximizes the advantages of synergistic energy supply is needed. Mathematical planning is a common method for the optimal design of IESs. In order to determine the optimal

operation strategy for the system, Feng et al. investigated the impact of the operation strategy on the system performance based on two different chiller configuration schemes [20]. Hawkes and Liu et al. used a linear programming approach to achieve optimization of the system's operating strategy and equipment configuration [21,22]. As for the IESs with more equipment, due to the large number of decision variables involved, which increases the difficulty of solving the model, intelligent optimization algorithms, such as particle swarm optimization (PSO) and non-dominated sorting genetic algorithm (NSGA), are usually used for optimization and determining solutions [23]. Besides, due to the large dependence of solar and wind energy on the geographic environment, the supply of PV and wind power is unstable and uncontrollable [7]. If PV and wind power are coupled in IES, the effects of uncertainty in energy supply and user demand need to be considered simultaneously [24,25]. Bacekovic and Dincer conducted a simulation study with an IES for renewable energy access, and the results showed that the system does have advantages in terms of improving energy efficiency [26,27]. Carpaneto et al. found that, when considering the uncertainty of renewable energy supply [28], it is difficult to achieve trade-offs between operating strategies and equipment capacity obtained with the optimization objectives of economy and energy efficiency. Most of the above work is limited to the coupling of single renewable energy sources coupling with IES, and the environmental impacts caused by them have not been further explored. Finally, economic efficiency is an important factor in determining whether the optimal design of a system is reasonable or not, and the minimum economic cost is often used as the objective function of the optimal design of IES in current studies. Deng et al. used a mothballing algorithm to optimize the combined heat and power generation (CHP) system with economic objective [29]. Lu and Afzali et al. explored the correlation between energy price and system equipment capacity [18,30]. However, the objective function in most of the current studies is relatively singleness, the parameters affecting the energy system are not considered comprehensively enough, and the correlation between natural gas prices and electricity prices is not discussed; therefore, the coupling relationship between them cannot be ignored [31,32]. Table 1 provides a brief summary of energy system related research.

Considering the main contents and limitations of the existing research on IES, this paper takes an industrial park in Xi'an, China, as a specific case and carries out a more comprehensive optimization design, evaluation, discussion, and analysis of its IES based on the comprehensive consideration of equipment capacity configuration, operation strategy, multi-indicator evaluation, and the impact of source-load uncertainty. The specific work is completed as follows:

- By establishing a linear programming model with the minimum annual total operating cost of IES as the objective function, the equipment capacity configuration and operation scheduling strategy of the IES system coupled with PV and wind power are optimally designed.
- A comprehensive evaluation framework including economic, energy efficiency, and environmental aspects is constructed to comprehensively evaluate the performance of IES.
- Taking the IES of an industrial park in Xi'an, China, as the specific case, a sensitivity analysis is used to explore the impact of various key parameters on equipment capacity and operation strategies.
- Considering the regional characteristics of the impacts of PV and wind power on IES, the Monte Carlo method is utilized to investigate the impacts of source-load uncertainty on the equipment configuration and output profile of IES in Xi'an area, China.

**Table 1.** A brief summary of energy system related research.

| Scenarios | Model | Advantage | Outlook |
|---|---|---|---|
| Residential building [12] | Maximum rectangle method (MRM) | This paper explores the benefits of using a hybrid-CCHP system instead of a basic-CCHP system. The solar collector orientation and type is optimized. | Choosing the best solar strategy to design a collector. |
| Energy community [13] | MRM, Particle Swarm Optimization (PSO) | The study combines hydrogen energy and thermal energy storage to streamline device configuration | The analysis of detailed thermodynamic energy flow. |
| Sea island [14–16] | Traversing method Branch-and-bound method | The study offered valuable insights into the integration of desalination with the CCHP system. | Multi-objective method is used to solve the conflict problem |
| Commercial region [18] | Mixed-integer linear programming model (MILP) | This project employs consistent energy demands and average seasonal weather conditions for IES design. | Focus on uncertainties in renewable energy sources and energy demands. |
| Zagreb [26] | EnergyPLAN (simulation study) | This article compares two approaches to achieve a 100% renewable energy system in a city: traditional and smart systems. | Impact of some primary factors on intermittent renewable energy production. |
| Hotel building [31] | Moth Flame Optimization algorithm | It provides a reference for the study of equipment operating under off-design performance conditions in IES. | The impact of key parameter settings on system and equipment performance. |
| Central business district [33] | Multi-objective genetic algorithm | Proposes a new CCHP system model that segments operating conditions and integrates the part-load performance of power generation unit. | Energy storage devices can be added to the energy system. |
| Industrial Park [34] | GA | Proposes an integrated method to optimize configuration and strategy of CCHP systems. | The study needs to incorporate multi-objective optimization thoroughly. |

The remainder of this paper is arranged as follows: Section 2 establishes a mathematical optimization model for integrated energy systems with the annual total cost as the objective. It also outlined the corresponding constraints and explained the evaluation criteria regarding economics, energy efficiency, and environmental aspects. Relevant parameters and data sources used in the computational case studies in this paper are introduced in Section 3. The experiment results and statistical analyses are presented in Section 4. Finally, Section 5 provides a summary.

## 2. Methodology

Due to the change in energy demand, the energy supply mode has gradually changed from the sub-supply energy system to the integrated energy system, which has mainly experienced the development stages of the traditional sub-supply system, the cogeneration system of cold, heat, and electricity, the integrated energy system coupled with renewable energy, and the integrated energy system with energy storage devices on this basis. The comparison between the traditional sub-supply energy system and the integrated energy system is shown in Figure 1. In Figure 1, "$\oplus$" represents the coupling node of various energy flows, and "$\ominus$" represents the separation node of various energy flows. The structure of the integrated energy system studied in this paper is shown on the left in in Figure 1.

The integrated energy system studied in this paper consists of three parts, namely, the power system, the gas system, and the energy storage system, which meet the cooling load, heat load, and electric load requirements of the users. The power system of the integrated energy system includes wind power generation subsystem, PV power generation subsystem, and power grid. The electrical energy demand within the conventional sub-supply energy system is entirely accommodated by the grid. Additionally, an electric chiller system is integrated to provide the required cooling load for users. The gas system comprises a natural gas pipeline network, gas turbines, and gas boilers. Waste heat boilers recover excess heat from the system, and an absorption chiller system utilizes waste heat

from the gas turbines to provide cooling load for end-users. The energy storage system consists of electrical energy storage subsystems and thermal energy storage subsystems, with the latter serving the heating load requirements of users.

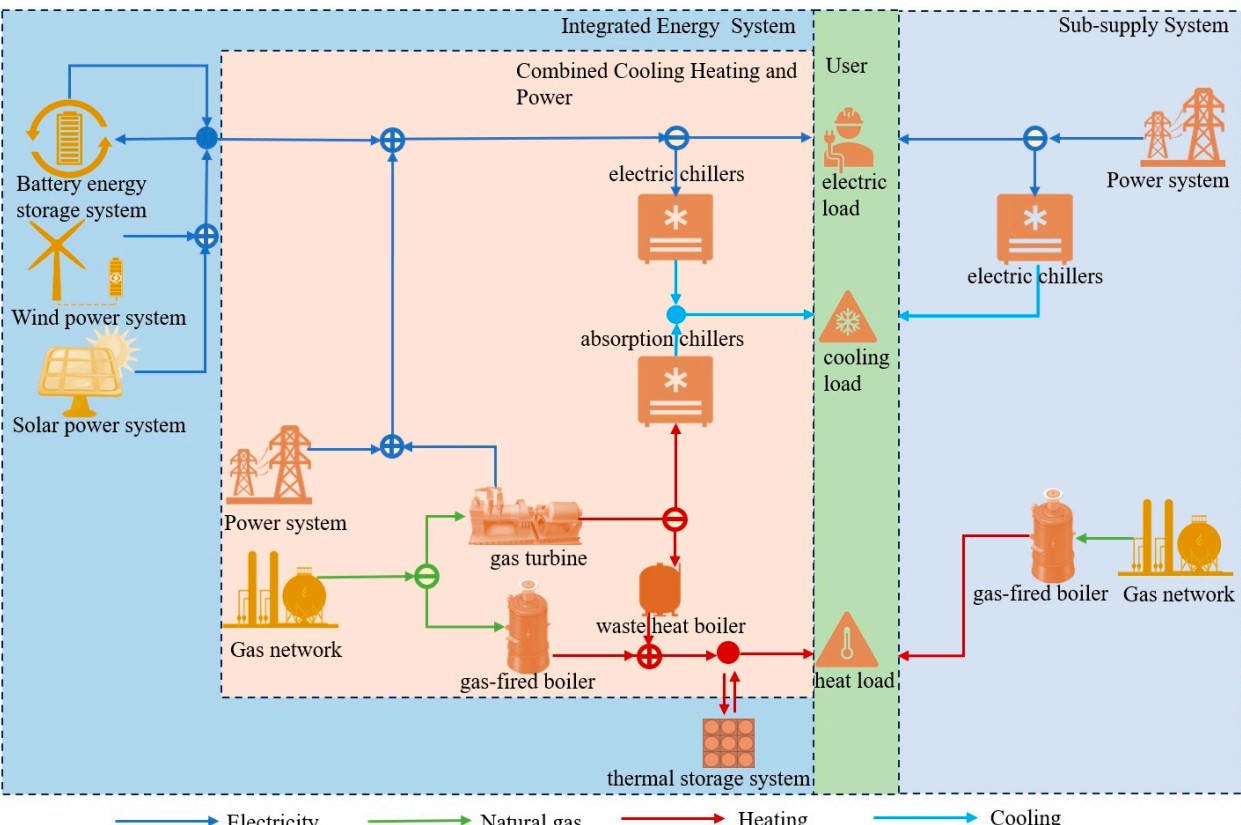

**Figure 1.** Comparison of sub-supply system and integrated energy system.

Wind power generation and PV power generation have significant uncertainties, so this paper limits their power generation proportion in the follow-up study.

*2.1. Optimization*

2.1.1. Objective Function

The development of an integrated energy system must rely on the law of the market, so its economy is one of the main motives for investors' choices [27]. Therefore, the optimization model in this paper chooses the lowest annual total cost (ATC) as the objective function to ensure that the integrated energy system brings higher revenue to the operator, as shown in Equation (1).

$$ATC = U \times I_{IES} + R_{IES} + M_{IES} - I_{\text{sale}} \qquad (1)$$

In the formula, U represents the investment recovery factor, which is employed to allocate the initial total investment cost annually, as computed in Equation (2). $I_{IES}$, $R_{IES}$, $M_{IES}$ and $I_{sale}$ denote the initial capital cost, annual operational cost, annual maintenance cost, and annual profit from selling excess electricity from the integrated energy system, as calculated according to Equations (3)–(6), respectively, for the integrated energy system.

$$U = \frac{i(1+i)^y}{(1+i)^y - 1} \qquad (2)$$

$$I_{IES} = \sum_{n=1}^{N} S_n \times c_n \tag{3}$$

$$M_{IES} = \sum_{m=1}^{12} \sum_{d=1}^{30} \sum_{h=1}^{24} \sum_{n=1}^{N} (O_n \times m_n)_{m,d,h} \tag{4}$$

In the formula, *i* is the annual interest rate, %; y is system lifetime, year; m, d, and n represent the month, day, and hour, respectively; N denotes the number of devices in the system; $S_n$ is the capacity of device n, kW; $c_n$ and $m_n$ are the initial investment cost per unit capacity and maintenance cost per unit output of equipment n, respectively, CNY/kW; and $O_n$ is the output of device n, kW·h.

The operational costs of the equipment primarily involve the consumption of natural gas and the purchase of electricity from the grid. According to the topological diagram, it can be determined that natural gas is supplied to the gas turbine and boiler while electricity is distributed to the electric refrigeration unit, users, and the battery system.

$$R_{IES} = \sum_{m=1}^{12} \sum_{d=1}^{30} \sum_{h=1}^{24} \left( \left( F_{GT}^{\text{grid}} + F_{GB}^{\text{grid}} \right) \times P_{\text{ng}} + \left( E_{EC}^{\text{grid}} + E_{\text{user}}^{\text{grid}} + E_{SB}^{\text{grid}} \right) \times P_e \right)_{m,d,n} \tag{5}$$

$$I_{sale} = \sum_{m=1}^{12} \sum_{d=1}^{30} \sum_{h=1}^{24} \left( E_{grid}^{GT} + E_{grid}^{PV} + E_{grid}^{WT} \right)_{m,d,h} \times P_{\text{sale}} \tag{6}$$

In the formula, $P_e$, $P_{ng}$, and $P_{sale}$ are the price of electricity purchased from the grid, the price of natural gas, and the price of electricity sold to the grid, respectively, CNY/kWh; $F_{GT}^{grid}$ and $F_{GB}^{grid}$ are the natural gas consumption of gas turbines and gas boilers, respectively, kWh; $E_{EC}^{grid}$, $E_{user}^{grid}$, and $E_{SB}^{grid}$ are the amounts of electricity of the electric refrigerator, the user, and the storage battery, respectively, kWh.

2.1.2. Equipment Constraints

The constraints of the integrated energy system's configuration and operational optimization model encompass user demand constraints and equipment constraints. Equipment constraints can be further categorized into equipment performance constraints, capacity constraints, and energy balance constraints. These constraints are derived from the objective conditions and limitations that must be met to ensure the system's proper operation. This paper involves seven key equipment components, and the following constraints need to be satisfied.

(1) Photovoltaic power generation system

The electrical power generation of the PV system can be considered a function of solar radiation intensity and environmental temperature. When the solar irradiance falls below the standard illumination conditions, the PV system does not operate at full capacity. The specific formula for performance constraints is shown in Equation (7).

$$P_{PV}(t) = \begin{cases} f_{PV} \times P_{PV,RP} \times \frac{I(t)}{I_{STC}} \times [1 - \delta(T_c(t) - T_{STC})], I(t) \leq I_{STC} \\ P_{PV,RP} \qquad , I(t) \leq I_{STC} \end{cases} \tag{7}$$

In equation, $P_{PV}(t)$ is the output power of the PV generator set at time t; $f_{PV}$ is the power derating factor for the PV unit; $P_{PV,RP}$ is rated power of unit; $I(t)$ is the intensity of solar radiation at time t; $I_{STC}$ is the radiation intensity of sunlight under standard lighting conditions, 1 kw/m$^2$; $\delta$ is the temperature coefficient of PV power generation units (negative value); $T_c(t)$ is the temperature of the PV cells at time t; $T_{STC}$ is the PV cell temperature under standard test conditions, and is typically set at 298.15 K (temperature in kelvin).

The destinations of the PV power generation system include the electric refrigeration unit, the storage battery system, and various users. Thus, the energy balance constraint for the PV power generation system is represented by Equation (8). Equation (9) presents the upper and lower bounds constraints for the electrical energy supply from the PV system.

$$E_{\text{user}}^{PV}(t) + E_{EC}^{PV}(t) + E_{ES}^{PV}(t) = E_{PV}(t) \tag{8}$$

$$E_{\text{user}}^{PV}(t) + E_{EC}^{PV}(t) + E_{ES}^{PV}(t) \leq E_{RP}^{PV}(t) \tag{9}$$

Per the mathematical equation, $E_{\text{user}}^{PV}(t)$ and $E_{EC}^{PV}(t)$ represent the electrical energy supplied by the PV power generation system to the users and the electric refrigeration unit, respectively; $E_{ES}^{PV}(t)$ represents the electrical energy stored in the battery by the PV power generation system; $E_{RP}^{PV}(t)$ is the rated power generation of the PV generation unit.

(2) Wind power generation system

The electricity generation of the wind power system depends on the variation in wind speed throughout the day, and the operational conditions of the wind turbine are determined by the cut-in wind speed, rated wind speed, and cut-out wind speed. The wind turbine remains inactive when the wind speed is below the cut-in wind speed or exceeds the cut-out wind speed. When the wind speed exceeds the rated wind speed but is below the cut-out wind speed, the motor operates at full load. When the wind speed is greater than the cut-in wind speed but less than the rated wind speed, the motor power generation constraint formula is shown in Equation (10). The energy balance constraint and upper/lower bounds constraints are represented by Equations (11) and (12).

$$P_{WT}(t) = \begin{cases} 0 & , v < v_{CI} \text{ or } v > v_{CO} \\ P_{WT,RP}(t) \times \left( \frac{v^3 - v_{CI}^3}{v_{RS}^3 - v_{CI}^3} \right), & v_{CI} < v < v_{RS} \\ P_{WT,RP}(t) & , v_{RS} < v < v_{CI} \end{cases} \tag{10}$$

In the formula, $P_{WT}(t)$ is the output power of the wind power generation; $P_{WT,RP}(t)$ is the rated power generation; $v_{CI}$, $v_{CO}$, and $v$ are the cut-in wind speed, cut-out wind speed, and actual wind speed, respectively.

$$E_{WT}(t) = E_{\text{user}}^{WT}(t) + E_{EC}^{WT}(t) + E_{ES}^{WT}(t) \tag{11}$$

$$E_{\text{user}}^{WT}(t) + E_{EC}^{WT}(t) + E_{ES}^{WT}(t) \leq E_{WT}(t) \tag{12}$$

In the formula, $E_{WT}(t)$ represents the electrical energy supplied by the wind power generation; $E_{\text{user}}^{WT}(t)$ and $E_{EC}^{WT}(t)$ represent the electrical energy supplied by the wind power generation system to the users and the electric refrigeration unit, respectively; and $E_{ES}^{WT}(t)$ represents the electrical energy stored in the battery by the wind power generation system.

(3) Gas turbine

The electricity generation and recoverable waste heat of the gas turbine depend on the consumed fuel quantity and equipment capacity. Consequently, the performance constraint model of the gas turbine can be formulated as shown in Equation (13). The energy balance constraint and upper/lower bounds constraints are represented by Equations (14) and (15).

$$P_{GT}(t) = V_{GT} \eta_{GT} LHV_{gas} \tag{13}$$

$$0 \leq P_{GT}(t) \leq P_{GT} \tag{14}$$

where $P_{GT}$ is the output power of the gas turbine at time t; $V_{GT}$ is the volume of natural gas consumed by the gas turbine; $\eta_{GT}$ is the conversion efficiency of gas turbine; $LHV_{gas}$ is the low calorific value of natural gas; and $P_{GT}$ is the rated power of the gas turbine.

$$E_{GT} = E_{\text{user}}^{GT} + E_{ES}^{GT} + E_{EC}^{GT} \tag{15}$$

$$Q_{WH}^{GT} = Q_{\text{user}}^{GT} + Q_{TSS}^{GT} + Q_{AC}^{GT} \tag{16}$$

In the formula, $E_{user}^{GT}$, $E_{EC}^{GT}$, and $E_{ES}^{GT}$ represent the electrical energy supplied by the gas turbine to the users and the electric refrigeration unit, and that stored in the battery, respectively. $Q_{\text{user}}^{GT}$, $Q_{TSS}^{GT}$, and $Q_{AC}^{GT}$ represent the thermal energy supplied to users and the absorption refrigeration machine, and that stored in the thermal energy storage device, respectively, all of which are recovered from the gas turbine's waste heat.

(4) Absorption chiller

The absorption refrigeration machine is a device that utilizes thermal energy to drive the refrigeration process, providing a cooling load to users without the need for a compressor. In the context of this study, the source of thermal energy is the recovered waste heat from the gas turbine, which enhances the system's energy efficiency. The refrigeration capacity is contingent upon the amount of waste heat; therefore, the refrigeration unit's operation strategy is associated with waste heat recovery. The performance constraint equation for the refrigeration machine is presented in Equation (17), and the upper and lower bounds constraints are formulated in Equation (19).

$$Q_{AC}^{cool}(t) = Q_{GT}(t)\eta_{AC}COP_{AC} \tag{17}$$

$$Q_{WH}^{GT} = \frac{P_{GT}(1 - \eta_{GT} - \eta_1)}{\eta_{GT}} \tag{18}$$

$$0 \leq P_{AC}(t) \leq P_{AC,RP} \tag{19}$$

In the formula, $Q_{WH}^{GT}$ represents the waste heat generated by the gas turbine; $\eta_{AC}$ is the efficiency of waste heat recovery; $Q_{AC}^{cool}$ is the cooling capacity of the absorption chiller; and $COP_{AC}$ is the coefficient of cooling. $\eta_1$ represents the coefficient of heat loss during gas turbine heat dissipation; $P_{AC}(t)$ is the cooling power of the absorption chiller at time t; and $P_{AC,RP}$ is the rated power of the absorption chiller.

(5) Gas boiler

The gas boiler within the integrated energy system utilizes natural gas as fuel, generating high-temperature flue gas through combustion. It then conducts heat exchange with a working medium across a heated surface to produce steam or hot water, subsequently supplied to users. In this study, the energy source for the boiler is natural gas, meaning that the heat generated by the boiler is primarily determined by the quantity of natural gas consumed. The upper and lower bound constraints for the gas boiler are defined in Equation (21).

$$H_{GB}(t) = V_{GB} \times LHV \times \eta_{GB} \tag{20}$$

$$0 \leq H_{GB}(t) \leq P_{GB} \tag{21}$$

In the equation, $H_{GB}(t)$ is the output power of the gas boiler at time t; $V_{GB}$ is the amount of natural gas consumed by the gas boiler; $\eta_{GB}$ is the conversion efficiency of gas boiler; and $P_{GB}$ is the rated power of the gas boiler.

(6) Electric chiller

In this study, the electric chiller unit in the system employs a compression refrigeration system, where the refrigeration capacity of the electric refrigeration unit is contingent upon the electrical energy consumption. The performance constraint is defined by Equation (22),

the energy balance constraint is represented by Equation (23), and the upper and lower bound constraints are formulated in Equation (24).

$$Q_{EC} = E_{EC} \times COP_{EC} \tag{22}$$

$$E_{EC} = E_{EC}^{GT} + E_{EC}^{PV} + E_{EC}^{WGE} + E_{EC}^{grid} + E_{EC}^{SB} \tag{23}$$

$$0 \leq Q_{EC} \leq P_{EC} \tag{24}$$

Per the mathematical equation, $Q_{EC}$ is the cooling capacity of the electric chiller and $E_{EC}$ is the electrical energy consumed by electric chiller. $E_{EC}^{grid}$ is the electrical power supplied to the electric chiller unit by the grid; $E_{EC}^{SB}$ is the electrical power supplied to the electric chiller unit by the electricity storage equipment; and $P_{EC}$ is the rated power of the electric chiller.

(7) Energy storage device

The energy storage devices within the integrated energy system consist of batteries and thermal storage tanks. Batteries convert electrical energy into chemical energy for storage while thermal storage tanks store energy using a thermal storage medium.

At any given moment, the charge and discharge power of the battery is related to the stored electrical quantity, with the energy storage device being a passive component, so the charge and discharge amount should be equal. The initial charge per day is set as equal to the charge at the 24th hour. The battery model is presented in Equation (25), and operational constraints are defined by Equation (26).

$$E_{t+1}^{SB} = E_t^{SB} + (\eta_{SB,chr} P_t^{SB,chr} - P_t^{SB,dis} / \eta_{SB,dis})\Delta t \tag{25}$$

$$\begin{cases} 0 \leq P_t^{SB,chr} \leq \varepsilon_t^{SB,chr} \lambda_{chr}^{SB} E_{cap}^{SB} \\ 0 \leq P_t^{SB,dis} \leq \varepsilon_t^{SB,dis} \lambda_{dis}^{SB} E_{cap}^{SB} \\ \varepsilon_t^{SB,chr} + \varepsilon_t^{SB,dis} \leq 1 \\ \delta_{low}^{SB} E_{cap}^{SB} \leq E_t^{SB} \leq \delta_{up}^{SB} E_{cap}^{SB} \end{cases} \tag{26}$$

where $E_{t+1}^{SB}$ represents the electric energy stored in the battery at time t + 1, and $P_t^{SB,chr}$ and $P_t^{SB,dis}$ denote the charging and discharging power of the battery at time t, respectively. $\eta_{SB,chr}$ and $\eta_{SB,dis}$ denote the charging and discharging efficiency of the battery, respectively. $E_{cap}^{SB}$ is the capacity of battery, and $\varepsilon_t^{SB,chr}$ and $\varepsilon_t^{SB,dis}$ represent the charge and discharge state indicator variables at time t, respectively, which are variables ranging from 0 to 1. $\varepsilon_t^{SB,chr} = 1$ indicates that the battery is in a charging state. $\lambda_{chr}^{SB}$ and $\lambda_{dis}^{SB}$ represent the maximum rate for battery charging and discharging power. $\delta_{up}^{SB}$ and $\delta_{low}^{SB}$ represent the upper and lower limit coefficients for the status value of the battery charge.

The model of the heat storage device is given in Equation (27), and the operation constraints of the heat storage device are given in Equation (28).

$$E_{t+1}^{TSS} = E_t^{TSS} + (\eta_{TSS,chr} Q_t^{TSS,chr} - Q_t^{TSS,dis} / \eta_{TSS,dis})\Delta t \tag{27}$$

$$\begin{cases} 0 \leq Q_t^{TSS,chr} \leq \varepsilon_t^{TSS,chr} \lambda_{chr}^{TSS} E_{cap}^{TSS} \\ 0 \leq Q_t^{TSS,dis} \leq \varepsilon_t^{TSS,dis} \lambda_{dis}^{TSS} E_{cap}^{TSS} \\ \varepsilon_t^{TSS,chr} + \varepsilon_t^{TSS,dis} \leq 1 \\ \delta_{low}^{TSS} E_{cap}^{TSS} \leq E_t^{TSS} \leq \delta_{up}^{TSS} E_{cap}^{TSS} \end{cases} \tag{28}$$

In equation, $E_{t+1}^{TSS}$ represents the thermal energy stored in thermal storage devices at time t + 1; $\eta_{TSS,chr}$ and $\eta_{TSS,dis}$ denote the charging and discharging efficiency of the

thermal storage device. $E_{cap}^{TSS}$ is the capacity of the thermal storage device, and $\varepsilon_t^{TSS,chr} = 1$ indicates that the thermal storage device is in a charging state.

(8) User demand constraints

The integrated energy system is designed to ensure the satisfaction of the system's cooling, heating, and electrical demand while achieving an efficient and reliable energy supply. The energy output from the equipment should, to a certain extent, exceed the users' load requirements. User demand constraints are given in Equations (29)–(32).

$$E_{PV}(t) + E_{WT}(t) + E_{GT}(t) - E_{SB}^{chr}(t) + E_{SB}^{dis}(t) + E_{grid}^{buy}(t) - E_{grid}^{sell}(t) \geq E_{user}(t) \qquad (29)$$

$$Q_{GB}(t) + Q_{user}^{GT}(t) + Q_{TSS}^{GT}(t) - Q_{chr}^{TSS}(t) + Q_{dis}^{TSS}(t) \geq Q_{user}(t) \qquad (30)$$

$$Q_{LBR}(t) + Q_{EC}(t) \geq Q_{user}^{cool}(t) \qquad (31)$$

$$Q_{user}(t) = \frac{Q_{heating}}{\eta_{heating}} + Q_{hotwater} \qquad (32)$$

In the formula, $Q_{heating}$ and $Q_{hotwater}$ are the user heat load and user hot water load, respectively.

### 2.2. Comprehensive Assessment Framework

The objective function is designed to optimize the economic performance of the integrated energy system in the previous section. However, in certain situations, optimizing economic performance may not necessarily translate to overall benefits or could even be detrimental to the system in other aspects. Therefore, this section proposes a comprehensive evaluation framework that considers economic, energy, and environmental impact factors to further assess the optimization results from multiple perspectives.

#### 2.2.1. Economic Indicators

The annual cost-saving rate (ACSR) is another economic indicator, similar to the ATC mentioned above. The ACSR represents the proportion of cost savings achieved by the integrated energy system compared to the comparison system relative to the total annual cost of the comparison system. ACSR is a relative metric for evaluating the economic benefits of the integrated energy system compared to the comparison system.

$$ACSR = \frac{ATC_{compare} - ATC_{IES}}{ATC_{compare}} \qquad (33)$$

#### 2.2.2. Energy Indicators

The primary energy utilization rate refers to the proportion of energy used that is directly converted into valuable energy during the energy conversion and utilization processes. In this context, the primary energy utilization rate represents the combined efficiency of the integrated energy system in this paper, and it is equal to the sum of various user loads divided by the total primary energy input into the system.

$$PEE = \frac{\sum\limits_{m=1}^{12} \sum\limits_{d=1}^{30} \sum\limits_{h=1}^{24} (E_{user} + H_{user} + R_{user})}{\sum\limits_{m=1}^{12} \sum\limits_{d=1}^{30} \sum\limits_{h=1}^{24} \left( F_{grid} + \frac{E_{grid}}{\eta_{grid}\eta_{trans}} \right)}$$

$$F_{grid} = F_{GB}^{grid} + F_{GT}^{grid}$$

$$E_{grid} = E_{user}^{grid} + E_{EC}^{grid} + E_{SB}^{grid} \qquad (34)$$

In this equation, $E_{user}$, $H_{user}$, and $R_{user}$ are the electrical load, thermal load, and cooling load of the user, respectively. $F_{GB}^{grid}$ and $F_{GT}^{grid}$ are the gas consumption of the boiler and gas

turbine, respectively. $E_{user}^{grid}$, $E_{EC}^{grid}$, and $E_{SB}^{grid}$ are the amount of electricity delivered from the grid to the user, the electric chiller, and the storage battery, respectively. $\eta_{grid}$ and $\eta_{trans}$ are the plant's power generation efficiency and the grid's transmission efficiency, respectively.

The energy utilization enhancement rate (EUER) refers to the percentage improvement in the primary energy utilization rate of the integrated energy system compared to the comparison system, reflecting the energy-saving advantages of the integrated energy system.

$$EUER = \frac{\eta_{IES} - \eta_{compare}}{\eta_{compare}} \times 100\% \tag{35}$$

### 2.2.3. Environmental Indicators

The excessive emission of greenhouse gases leads to global warming, so carbon dioxide emission (CDE) is an important indicator to measure the impact of energy systems on the environment.

$$CDE = \left( \mu_E E_{grid} + \mu_F F_{grid} \right) \tag{36}$$

where $E_{grid}$ is the electricity purchased from the grid, and $\mu_E$ and $\mu_F$ are the conversion factors of carbon dioxide for energy production.

The carbon dioxide emission reduction rate (CDRR) refers to the proportion by which the integrated energy system reduces carbon dioxide emissions compared to the comparison system.

$$CDRR = \frac{CDE_{compare} - CDE_{IES}}{CDE_{compare}} \tag{37}$$

## 3. Case Study

This study employs an industrial park in Xi'an as the research scenario and uses data collected from the park's energy management center. Four representative days, each representing one of the four seasons, are selected for optimization calculations. The park previously utilized a combined cooling heating and power (CCHP) system, and to enhance overall energy utilization efficiency, plans are in place to integrate PV and wind energy generation units. Considering the annual operating strategy as a decision variable, the coupled wind-solar energy storage integrated energy system involves over 17,000 optimization variables. To address the challenges mentioned above, this study utilizes data from these four representative days as load and meteorological data sources, thereby reducing data dimensionality.

User load is a critical parameter in the optimization model, constraining the minimum supply levels of various energy sources. The 24 h load data for four representative days in winter, spring, summer, and autumn are illustrated in Figure 2. Typical days are typically the statistical results of a certain parameter over a past period, such as temperature, load demand, solar radiation, wind speed, etc. These selected days represent the parameter variations under different seasons and weather conditions. The choice of typical days primarily aims at preserving crucial insights into the performance of a system. This is because certain patterns and behaviors repeat cyclically in energy systems. Opting for representative typical days enables the capture of these cyclic patterns while concurrently reducing the scale and complexity of computations. Moreover, researchers recommend using typical days for relevant investigations in IES [33–35]. Summer and winter represent the cooling and heating seasons, respectively, with significantly higher cooling and heating demands than autumn and spring. The annual electricity demand remains relatively stable. During non-working hours, from 9:00 p.m. to 6:00 a.m., only the essential electrical load for maintaining system operation is present, with no other user demand.

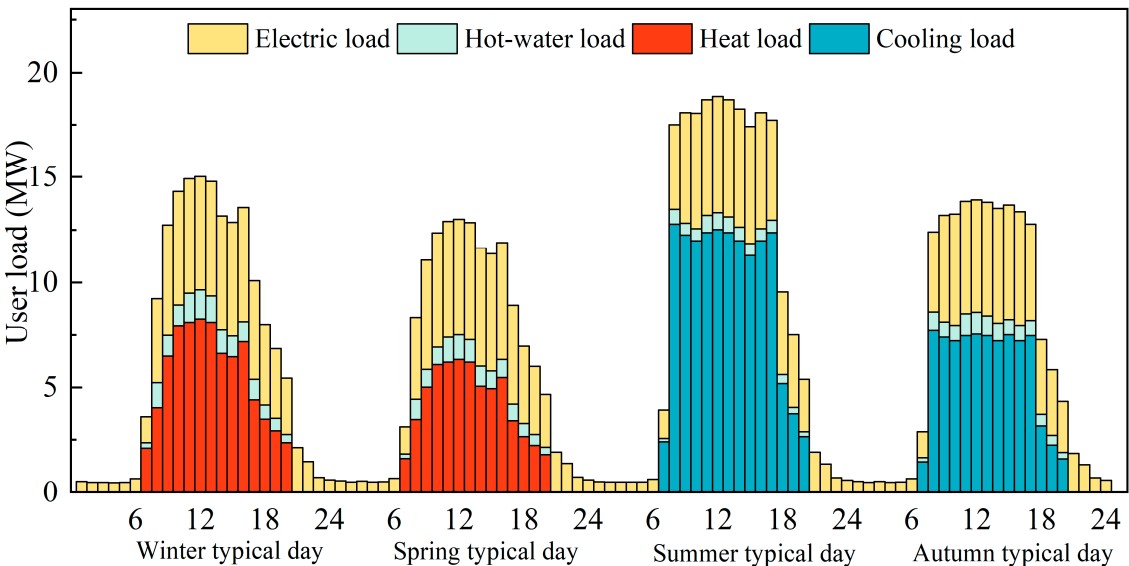

**Figure 2.** Load on four typical days.

The typical daily wind speeds and solar radiation intensity for the region where the industrial park is located are shown in Figure 3. As evident from the figure, the solar radiation intensity remains relatively consistent across the four representative days, while wind speeds exhibit seasonal variations.

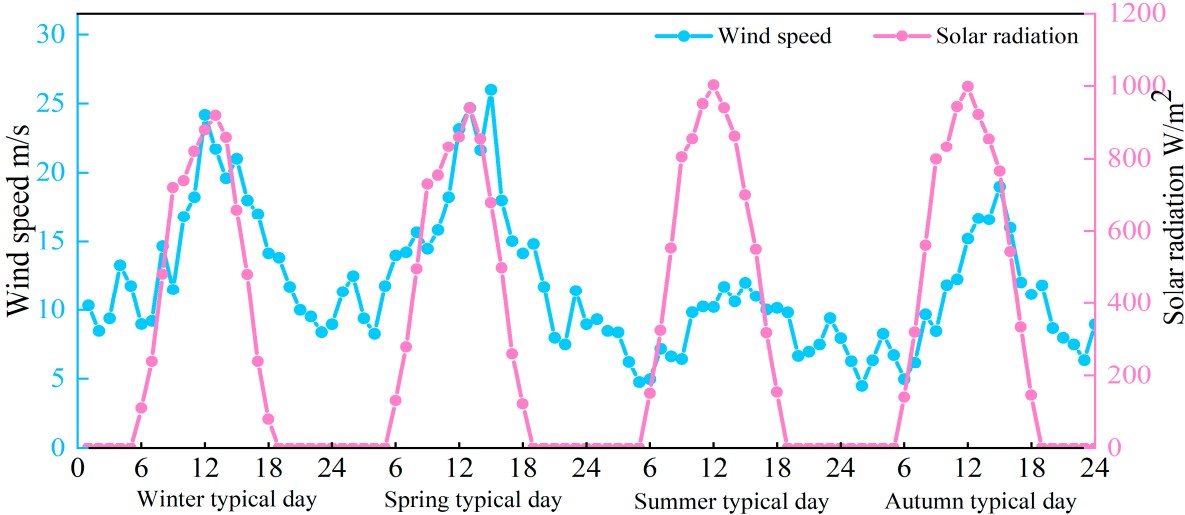

**Figure 3.** Four typical days of light intensity and wind speed.

*Parameters Setting and Optimization*

Optimization is performed based on the model established in the previous section to obtain the optimal equipment capacities and operational strategies for the integrated energy system with renewable energy integration. When determining the optimal equipment capacities, the capacities of the system's devices are considered as decision variables to be optimized. The results are then used as known parameters for optimal operational strategies.

For the established model, the following assumptions are established:

- Using the user load requirements for four representative days to represent the annual user load demand helps reduce the optimization dimension of the established model, i.e., the number of decision variables;

- The efficiency of the equipment remains constant, and the output of each device remains constant within an optimization time frame to ensure the solution speed and accuracy of the optimization model.

From the above assumptions, it can be seen that the optimization model adopted in this study is a linear programming (LP) model. Solving large-scale optimization problems has apparent advantages over the nonlinear programming (NLP) model in terms of global optimization and solution speed [36]. Although NLP models are considered closer to the actual operation of the device when solving planning problems [28,37], the difficulty of solving the problem has increased dramatically [38]. Therefore, a more efficient LP model is adopted in this paper.

The parameter values for the optimization model and evaluation criteria are provided in Tables 2 and 3. Technical parameters are presented in Table 2, while economic parameters are contained in Table 3. The computational analysis of the constructed model was carried out using MATLAB in this study.

**Table 2.** Technical parameters required for system evaluation.

| Item | Type | Parameter |
|---|---|---|
| electrical efficiency of gas turbine (%) | $\eta_{GT}$ | 0.35 |
| thermal efficiency of boiler (%) | $\eta_{GB}$ | 0.85 |
| refrigeration coefficient of absorption refrigerator | $COP_{AC}$ | 1.2 |
| refrigeration coefficient of electric refrigerator | $COP_{EC}$ | 4.8 |
| waste heat recovery rate of gas turbine [39] | $\eta_{re}$ | 0.8 |
| cut-in wind speed (m/s) | $v_{in}$ | 3 |
| cut-out wind speed (m/s) | $v_{out}$ | 20 |
| rated wind speed (m/s) | $V_{rated}$ | 10 |
| rated solar radiation intensity (Ix) | $I_{rated}$ | 1000 |
| electrical efficiency of power plant | $\eta_{grid}$ | 0.40 |
| transmission efficiency of electric grid [39] | $\eta_{trans}$ | 0.92 |
| emission factor—natural gas (g/kWh) [40] | $\mu_F$ | 220 |
| emission factor—power plant (g/kWh) [41] | $\mu_E$ | 600 |
| ambient temperature (K) | $T_{envir}$ | 273.15 |

**Table 3.** Economic parameters required for system evaluation. (CNY1 = \$0.15).

| Item | Type | Parameter |
|---|---|---|
| Designed life of system (Year) | y | 20 |
| Annual interest rate (%) | i | 4.2 |
| Initial cost per unit capacity of gas turbine (CNY/kW) | $c_{GT}$ | 6500 |
| Initial cost per unit capacity of absorption refrigerator (CNY/kW) | $c_{AC}$ | 1200 |
| Initial cost per unit capacity of electric refrigerator (CNY/kW) | $c_{EC}$ | 1000 |
| Initial cost per unit capacity of boiler (CNY/kW) | $c_{GB}$ | 900 |
| Initial cost per unit capacity of photovoltaics system (CNY/kW) | $c_{PV}$ | 5000 |
| Initial cost per unit capacity of wind turbine (CNY/kW) | $c_{WT}$ | 8000 |
| Initial cost per unit capacity of storage battery (CNY/kWh) | $c_{SB}$ | 6000 |
| Initial cost per unit capacity of thermal storage system (CNY/kWh) | $c_{TSS}$ | 6000 |
| Unit output maintenance cost of gas turbine (CNY/kWh) | $m_{GT}$ | 0.025 |
| Unit output maintenance cost of absorption refrigerator (CNY/kWh) | $m_{AC}$ | 0.015 |
| Unit output maintenance cost of electric refrigerator (CNY/kWh) | $m_{EC}$ | 0.015 |
| Unit output maintenance cost of boiler (CNY/kWh) | $m_{GB}$ | 0.015 |
| Unit output maintenance cost of photovoltaics system (CNY/kWh) | $m_{PV}$ | 0.025 |
| Unit output maintenance cost of wind turbine (CNY/kWh) | $m_{WT}$ | 0.015 |
| Unit output maintenance cost of storage battery (CNY/kWh) | $m_{SB}$ | 0.025 |
| Unit output maintenance cost of thermal storage system (CNY/kWh) | $m_{TSS}$ | 0.025 |
| Gas prices (CNY/kWh) | $p_g$ | 0.35 |
| Purchase price of electricity (CNY/kWh) | $p_p$ | 0.85 |
| Sale price of electricity (CNY/kWh) | $p_s$ | 0.55 |

## 4. Experimental Results and Discussion

Electricity prices, natural gas prices, thermoelectric ratios, feed-in tariffs, and the uncertainty of renewable energy sources are all critical parameters influencing the operation of IESs. In this section, to investigate the impact of these parameters on system equipment configuration and operational strategies, we first solve for the optimal configuration and operational strategies of the system. Then, we employ sensitivity analysis to examine variations in system configuration when key parameters fluctuate. Finally, we utilize Monte Carlo simulations to assess the influence of renewable energy source uncertainty on system operation.

### 4.1. Optimal Device Capacity of the System

Optimization designs are conducted for CCHP, renewable energy integrated energy systems (REIES), and energy storage integrated energy systems (ESIES), determining the optimal equipment capacities as depicted in Figure 4. CCHP comprises four components: a gas turbine, an electric chiller, an absorption chiller, and a boiler. REIES builds upon CCHP by incorporating PV and wind power generation systems, while ESIES further extends REIES by adding battery and thermal energy storage.

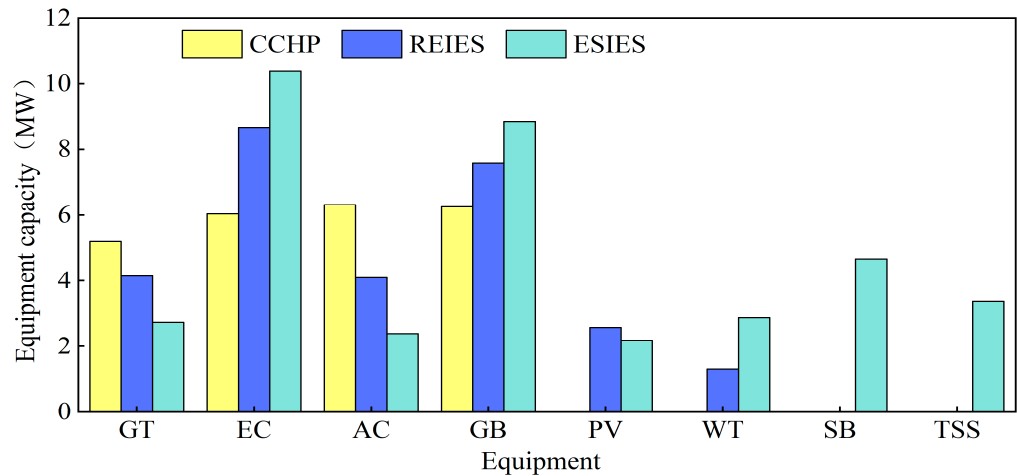

**Figure 4.** Optimal device capacity for the three systems.

The capacity of PV and wind power generation units exhibits distinct variations after the incorporation of energy storage systems. This change arises from PV systems operating only when there is sufficient sunlight. In contrast, wind power systems are unaffected by sunlight and can operate around the clock, resulting in a noticeable increase in capacity following the integration of energy storage systems.

Simultaneously, it can be observed that, with the integration of renewable energy sources and energy storage systems, there is a similar decreasing trend in the capacities of gas turbines and absorption chillers. Conversely, the capacities of electric chillers and gas boilers increase. This phenomenon arises because, most of the time, the thermal-to-electric ratio at the user end exceeds that of the gas turbine, which means more heat than electricity. Adding renewable energy sources satisfies some electric loads through renewable energy generation, exacerbating the imbalance and reducing the operational time of gas turbines. It is evident that, in systems with a surplus of electricity relative to heat, renewable energy generation units and batteries can drive the thermal–electric imbalance toward equilibrium. Under this condition, the capacities of gas turbines and absorption chillers are enhanced.

An additional electrical load is introduced on top of the existing thermal load to validate the abovementioned conjecture, resulting in a surplus of electricity relative to heat. The equipment capacities for the three systems are then recalculated, as depicted in Figure 5. It is evident that the incorporation of renewable energy sources does indeed increase the

capacities of gas turbines and absorption chillers, thus confirming the correctness of the conjecture above. However, after integrating energy storage systems, the capacities of gas turbines and absorption chillers decrease.

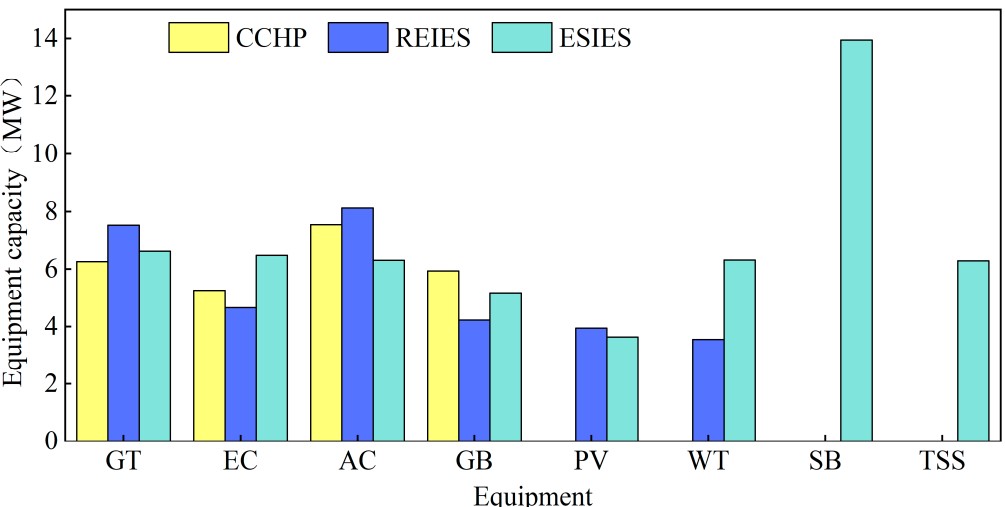

**Figure 5.** Optimal equipment capacity for each system after increasing the electrical load.

This phenomenon arises because the energy storage system has the capacity to consume renewable energy sources. With the augmentation of the electrical load, the capacity of the battery has expanded to three times that of the original electrical load condition, and the thermal storage system's capacity has doubled. This observation suggests that energy storage systems perform better under higher electrical load conditions.

The output conditions of ESIES equipment are illustrated in the accompanying Figure 6. It is evident from the graph that, when ESIES provides energy to users, the thermal load is predominantly carried by the gas boiler. In contrast, the electric chiller plays a primary role in meeting the cooling load. The contribution of the gas turbine remains relatively modest. With regard to power supply, during the winter and spring seasons, the gas turbine accounts for approximately 50% of the electricity supply. However, a significant amount of electricity must be procured from the grid during the summer and autumn, caused by increased cooling load and elevated power demand, particularly during peak cooling load periods. Consequently, the share of power generation by the gas turbine decreases. The PV system contributes to the daytime power supply only, while the wind power system and the grid jointly generate nighttime power.

At 14:00 on the typical autumn day, the gas turbine ceases operation because there is an abundant supply of wind and solar energy during this period, and electricity prices are at equilibrium. This leads to an imbalance of gas turbine energy supply state, making it economically less competitive.

The energy storage trends of the battery and thermal storage system over time are shown in Figure 7. From the graph, it can be observed that, during typical days in spring and winter, the trends in energy storage for the thermal storage system and the battery are similar. However, during typical days in summer and autumn, the energy storage behavior becomes more complex, with increased charging and discharging frequencies, which is related to the higher electricity consumption during the summer and autumn seasons.

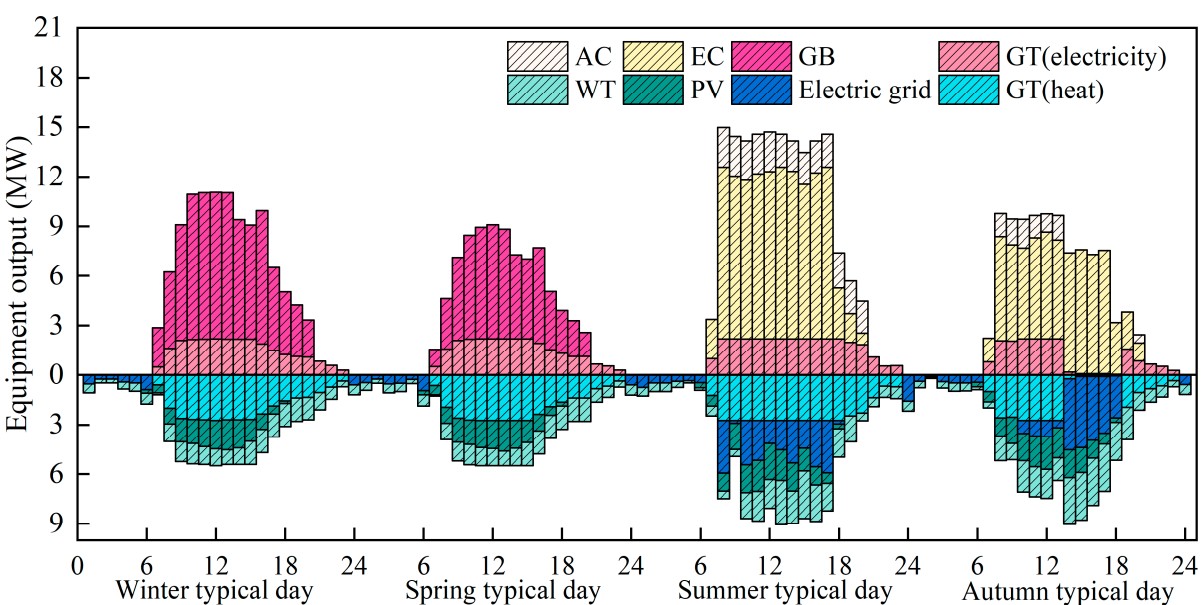

**Figure 6.** The output of each device in ESIES.

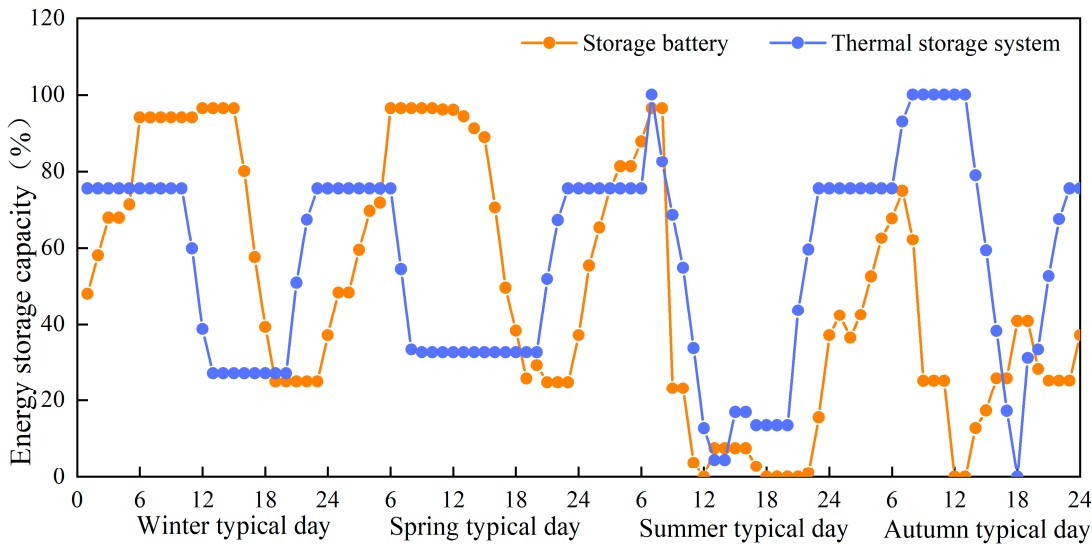

**Figure 7.** Changes in energy storage of batteries and heat storage tanks.

### 4.2. Research on Key Parameters

#### 4.2.1. Impact of Energy Prices on System Operation

Energy prices are influenced by macroeconomic regulations, and they exhibit relatively low volatility. Therefore, sensitivity analysis methods are employed for research. ESIES and REIES incorporate renewable energy sources, which mitigate the impact of price fluctuation. Hence, the primary focus of this section is on the CCHP system.

From the previous analysis, it is evident that the electricity load of the user is shared between the gas turbine and the grid. Therefore, with a constant electricity load, a game exists between these two sources. The outcome of this game is determined by the relative levels of the prices of these two energy sources. Hence, it is essential to consider both energy prices simultaneously rather than analyzing them independently.

The principle of solving linear programming problems using the simplex method involves traversing the edges of multidimensional polyhedra in search of the vertices that optimize the objective function. When there are minor fluctuations in two energy prices, these changes are insufficient to alter the position of the optimal vertex, and, as a result,

the optimization outcome remains unchanged. This phenomenon leads to the appearance of multiple plateaus in the solution space. While linear models may exhibit this step-like behavior, they still provide an overall representation of variations in the research variables.

Therefore, in this section, the optimization objective is to minimize the annual total cost while studying the impact of energy prices on the system. The computational results are shown in Figure 8. The initial electricity price is 0.85 CNY/kWh, and the natural gas price is 0.35 CNY/kWh. In the sensitivity analysis, both prices fluctuate within a range of $\pm 40\%$. The electricity price fluctuates between 0.5 to 1.2 CNY/kWh, while the natural gas price fluctuates between 0.2 to 0.5 CNY/kWh.

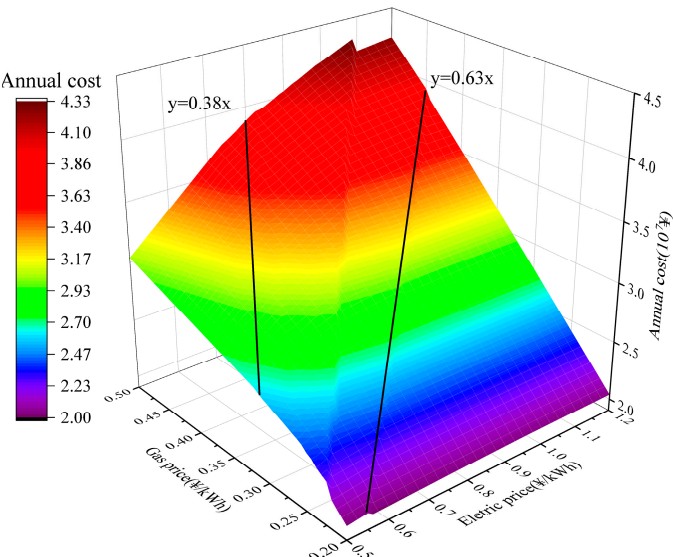

**Figure 8.** Total annual cost of the system as a function of electricity and gas prices.

Figure 8 shows that, for a given natural gas or electricity price, the extent to which annual system costs are influenced by the other one's energy price varies. When y > 0.63x, fluctuations in electricity prices have a more significant impact on the annual total cost. Conversely, when y < 0.38x, fluctuations in gas prices have a more significant impact on the annual total cost. When 0.38x < y < 0.62x, both energy prices have a roughly equal impact on the annual total cost. In other words, when electricity prices are more than 2.63 times higher than natural gas prices, increases in electricity prices have almost no effect on the annual system cost. Conversely, when natural gas prices are more than 0.63 times higher than electricity prices, increases in natural gas prices have a minimal impact on the annual system cost. This also reflects the asymmetric impact of the two energy prices on system costs.

The system efficiency and carbon dioxide emissions vary with electricity and natural gas prices, as shown in Figure 9. It can be observed that lower natural gas and higher electricity prices lead to a state of high efficiency and low carbon dioxide emissions in the system. This is because lower natural gas prices increase the utilization of gas turbines, allowing for better exploitation of the energy-cascaded utilization of co-generation units. The trends in the variation of both prices show some similarities and exhibit clear step-like patterns. When the set natural gas and electricity prices are within highly fluctuating regions, even slight changes in these prices can result in significant fluctuations in efficiency and carbon dioxide emissions.

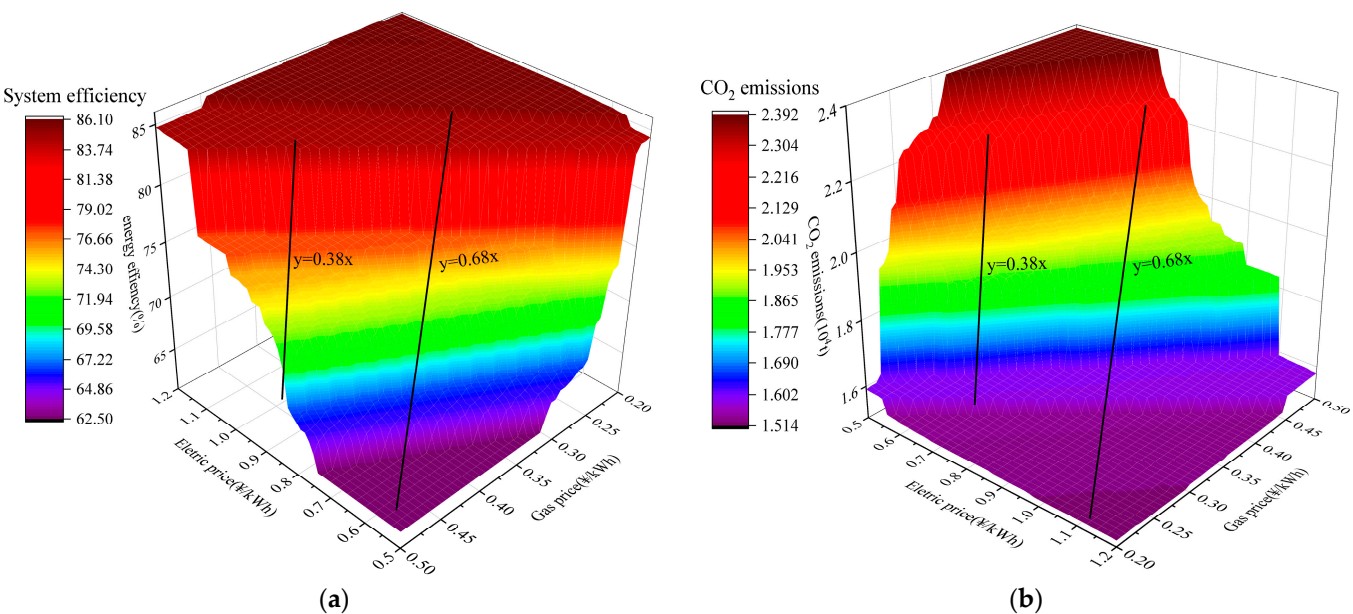

(**a**)  (**b**)

**Figure 9.** (**a**) Total annual cost of the system as a function of electricity and gas prices (**b**) Changes in $CO_2$ emissions as a function of electricity and gas prices.

### 4.2.2. Impact of Energy Prices on Capacity of Equipment

The variation in equipment capacity with electricity and natural gas prices is shown in Figure 10. Overall, as electricity prices increase and natural gas prices decrease, the capacity of gas turbines and absorption chillers increases, while the capacity of boilers and electric chillers decreases. There are similarities in the variations of gas turbines and gas boilers, as well as electric chillers and absorption chillers. This phenomenon arises due to competition among equipment that provides the same type of energy. Importantly, fluctuations in natural gas and electricity prices within a specific range do not lead to significant changes in equipment capacity, and this fluctuation range occurs when $y < 0.47x$.

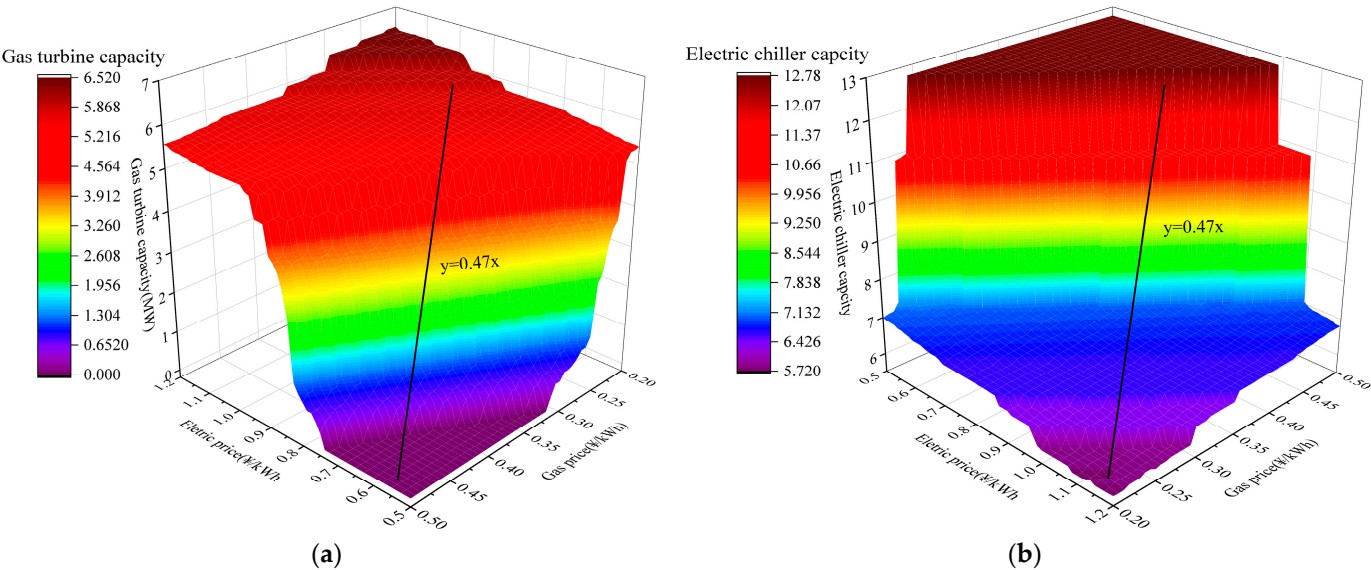

(**a**)  (**b**)

**Figure 10.** *Cont.*

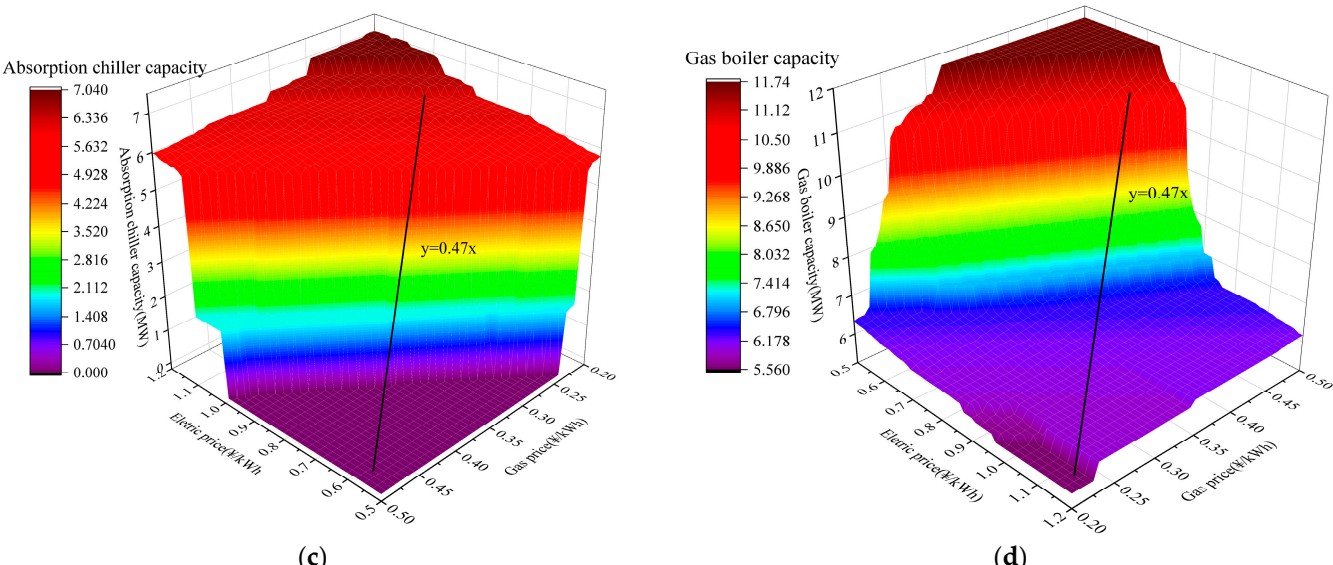

**Figure 10.** Changes in equipment capacity as a function of energy prices. (**a**) The capacity of the gas turbine as a function of electricity and gas prices. (**b**) The capacity of the electric chiller as a function of electricity and gas prices. (**c**) The capacity of the absorption chiller as a function of electricity and gas prices. (**d**) The capacity of the gas boiler as a function of electricity and gas prices.

Through comparing the capacity variations of the four devices, it can be observed that the capacity of gas turbines and absorption chillers can decrease to 0 in extreme cases. However, throughout the entire range of fluctuations in electricity and natural gas prices, the equipment capacity of gas boilers and electric chillers remains at a minimum of around 6MW. This indicates that, to ensure the economic viability of the system, the auxiliary function of these two types of equipment is indispensable.

### 4.2.3. Impact of Feed-in Tariffs on System Operation

The current policy regarding the on-grid electricity price is unclear, so analyzing its impact on system capacity is of research significance. The design calculates the equipment capacity when the electricity recovery price varies between 0.1 CNY and 0.9 CNY. It can be seen that, when it is less than 0.4 CNY, the equipment capacity hardly changes, and the electricity buyback price generally does not exceed 0.68 CNY. It is not meaningful to study when the buyback price is significantly higher than the price of purchasing electricity from the grid.

From Figure 11, it can be observed that, when the electricity buyback price is in the range of 0 CNY to 0.5 CNY, the equipment capacity hardly changes. This is because the electricity buyback price is too low, and the profit from selling electricity is insufficient to induce changes in the system's equipment capacity. However, when the price increases to 0.62, there is a significant change in equipment capacity, and afterward, the change in capacity levels off. The capacity of the electric refrigeration unit and gas boiler exhibits a similar trend, decreasing with an increase in electricity price. In contrast, the capacity of the gas turbine and absorption refrigeration unit increases with the price rise. This is because, as the electricity buyback price increases, the selling offsets a significant portion of the operating and maintenance costs of the gas turbine, making the cost of electricity generation by the gas turbine lower than the cost of purchasing electricity from the grid. A similar trend in the capacity of the absorption refrigeration unit is due to its capacity dependence on the gas turbine's capacity.

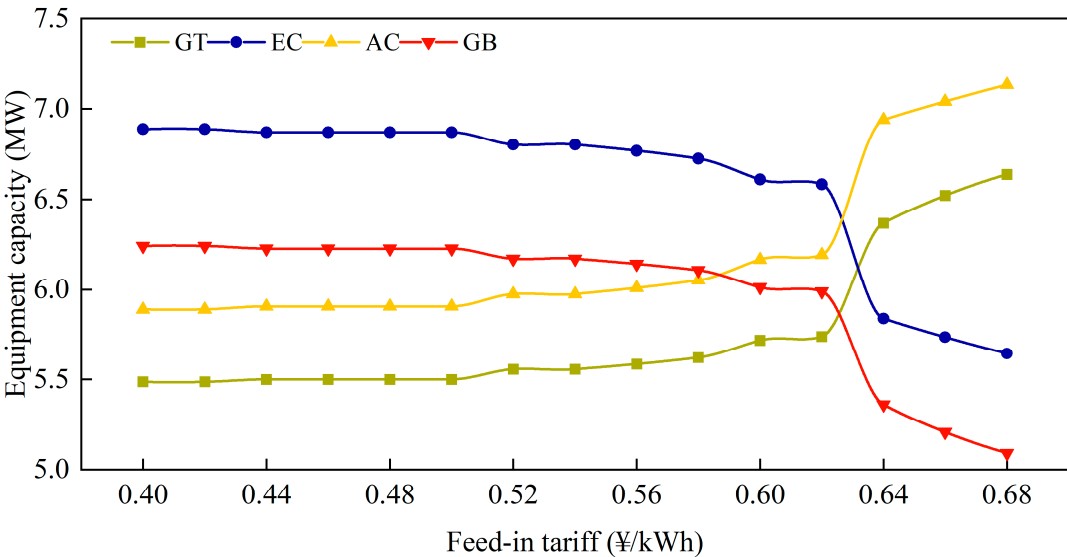

**Figure 11.** The impact of the feed-in tariff on equipment capacity.

It can be observed that, without considering the possibility of the feed-in tariff exceeding the grid electricity selling price, as the feed-in tariffs increases, the capacity of the gas turbine continues to increase, the absorption refrigeration unit's capacity increases to match the user's maximum cooling load, and the capacity of the boiler and electric refrigeration unit tends to approach zero. This is because selling electricity can generate profits for the system when the electricity buyback price is sufficiently high.

Currently, without subsidies, the electricity buyback price is below 0.6, meaning that the variation in electricity buyback prices has a limited impact on the equipment capacity of the system, with less than a 5% effect.

### 4.2.4. Impact of Thermoelectric Ratios on System Operation

In an integrated energy system, there are two parameters related to the thermal–electric ratio: the prime mover thermal–electric ratio and the load thermal–electric ratio. The prime mover thermal–electric ratio refers to the ratio of heat production to electricity generation by the prime mover. In contrast, the load thermal–electric ratio refers to the ratio of the sum of user heating load and cooling load conversion to electricity load. The proximity of these two thermal–electric ratios reflects the energy supply–demand relationship between the system and the users. The results are shown in Figure 12 by varying the user load and calculating the changes in evaluation metrics under different thermal–electric ratios. From the graph, it can be observed that all three evaluations metrics reach their peak values when the load thermal–electric ratio is equal to 1. The evaluation metrics show a decreasing trend as the thermal–electric ratio continues to increase or decrease.

The reason for the increasing trend of evaluation metrics with the variation of thermal–electric ratio is that, when the user heating load and electricity load differ significantly, the proportion of energy supplied by the prime mover decreases, limiting the energy cascade utilization. At the same time, the required prime mover capacity of the system gradually decreases, and the system tends toward a distributed energy supply. It can be predicted that, when the thermal–electric ratio approaches infinity or zero, the evaluation metrics will approach zero as well. The above analysis indicates that IESs do not always have advantages under all operating conditions. They only have significant advantages when a match in the thermal–electric ratio of supply and demand energy is achieved. In other words, when designing the system, the primary factor determining the system's operating mode is the composition of the user load, specifically the thermal–electric ratio.

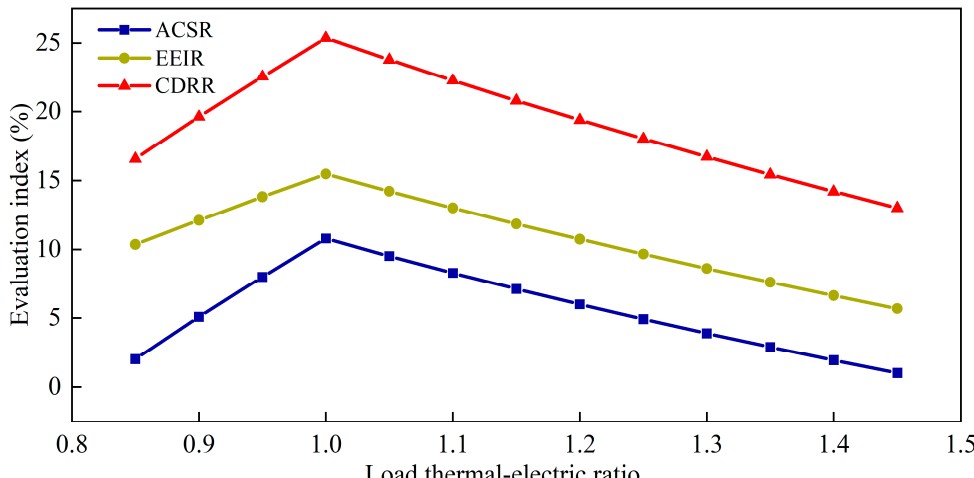

**Figure 12.** Influence of load thermal–electric ratio on three evaluation indexes.

To verify whether the system gradually approaches the SP system due to changes in the thermal–electric ratio, the calculation of the variation of equipment capacity is conducted with the thermal–electric ratio of the user load. The results are shown in the Figure 13. From this graph, it can be observed that the capacities of the gas turbine and absorption chiller exhibit a similar changing trend as the evaluation metrics, confirming the analysis that energy cascade utilization trends are limited by the thermal–electric ratio. However, as the thermal–electric ratio increases, the equipment capacity of electric chillers and absorption chillers does not increase as expected. This is mainly because changing the load thermal–electric ratio also alters the total user load.

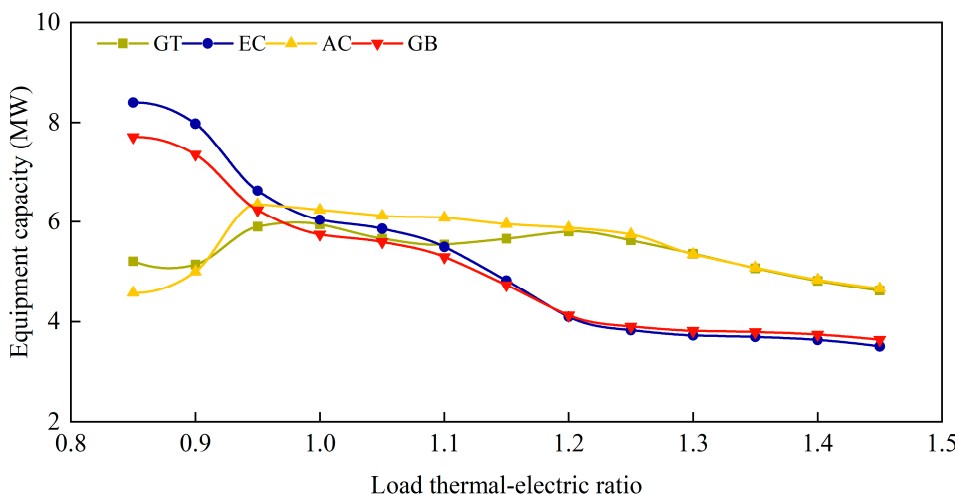

**Figure 13.** Change of equipment capacity with thermal–electric ratio.

### 4.2.5. Impact of Renewable Energy Uncertainty on System Operation

In practical real-life scenarios, user loads, wind speeds, and solar radiation values exhibit a certain degree of randomness. Simulating the stochastic fluctuations of the above data using the Monte Carlo method is necessary to study the impact of this uncertainty on the integrated energy system.

### User Load Probability Model

Several studies have already indicated that user load data follows a normal distribution. Field measurements have found that approximately 95% of the actual user load data falls within ±20% of the mean value [42]. This result has been widely utilized in research

regarding the uncertain. In this study, the data mentioned above are employed, and the probability density function for user load is represented as Equation (38).

$$f(D) = \frac{1}{\sqrt{2\pi}\sigma_D} \exp\left[-\frac{(D-\mu_D)^2}{2\sigma_D{}^2}\right] \tag{38}$$

In the equation, $D$ represents the user load, which can be cooling, heating, or electricity. $\mu_D$ represents the mean value of the corresponding load, and $\sigma_D{}^2$ represents the variance of the corresponding load.

Wind power generation depends on the wind turbine capacity and wind speed, and the uncertainty in wind speed leads to uncertainty in the wind power generation system's output. Several studies have indicated that the wind speed distribution for different days simultaneously follows a Weibull distribution. The probability density function of wind speed is shown in Equation (39). The shape and scale parameters of the Weibull distribution are region-specific.

$$f(v) = \frac{k}{c}\left(\frac{v}{c}\right)^{k-1} \exp\left[-\left(\frac{v}{c}\right)^k\right] \tag{39}$$

$$\mu_v = c\Gamma\left(1 + \frac{1}{k}\right) \tag{40}$$

$$k = \left(\frac{\mu_v}{\sigma_v}\right)^{-1.086} \tag{41}$$

In the equation, $v$ is the actual wind speed, $k$ is the shape parameters of Weibull distribution, $c$ is the scale parameters of the Weibull distribution [43], $\Gamma(\cdot)$ is the Gamma function [44], $\mu_v$ is the mean of a wind speed, and $\sigma_v$ is the standard deviation of the wind speed.

The electricity generation of PV systems depends on the capacity of the PV system and the local solar radiation intensity. The uncertainty in solar radiation intensity gives rise to uncertainty in PV system output. Solar radiation intensity follows a Beta distribution [43], and the probability density function of its random distribution is shown in Equation (42). The shape and scale parameters of the Beta distribution can be expressed using Equation (43) and Equation (44).

$$f(I) = \frac{\Gamma(\alpha+\beta)}{\Gamma(\alpha)+\Gamma(\beta)}\left(\frac{I}{I_{\max}}\right)^{\alpha-1}\left(1 - \frac{I}{I_{\max}}\right)^{\beta-1} \tag{42}$$

$$\alpha = \left(\frac{1-\mu_I}{\sigma_I{}^2} - \frac{1}{\mu_I}\right)\mu_I{}^2 \tag{43}$$

$$\beta = (1-\mu_I)\left(\frac{\mu_I(1-\mu_I)}{\sigma_I{}^2} - 1\right) \tag{44}$$

Per the mathematical equation, $I$ is the solar radiation intensity, $\alpha$ is the shape parameters of Beta distribution, $\beta$ is the scale parameters of Beta distribution, $\mu_I$ is the mean of solar radiation intensity, and $\sigma_I$ is standard deviation of solar radiation intensity.

Monte Carlo Simulation

The user load, wind, and PV probability models characterize the six-dimensional multivariate normal distribution of electricity, cooling, heating, hot water, wind speed, and solar radiation intensity for each hour of a typical day. Based on this six-dimensional multivariate normal distribution, 1000 sets of data with six variables each are generated using a random number generator for obtaining convergent results.

The Monte Carlo simulation process for studying renewable energy uncertainty is illustrated in Figure 14. This section randomly generates 1000 input data sets based on

the probability density functions. These data are then used with the optimization model discussed earlier to solve for equipment capacities and operational strategies, analyzing the impact of input uncertainty on the operation of the integrated energy system.

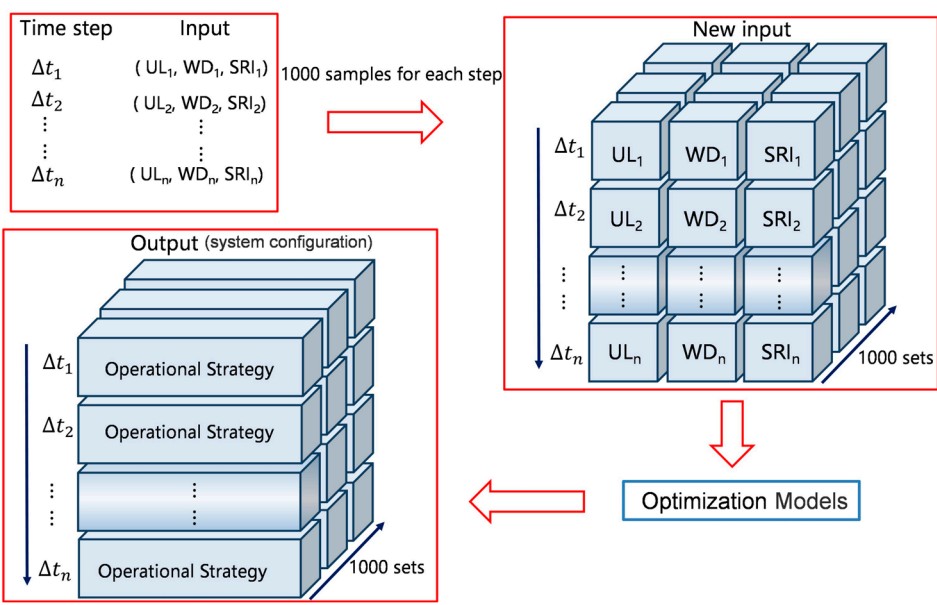

**Figure 14.** Flow chart of the Monte Carlo method.

The uncertainty in user load and renewable energy inputs leads to fluctuations in the optimal capacities. The fluctuation of the optimal capacity of each device affected by the uncertainty of the source load is shown in Figure 15. The CCHP system is only affected by the uncertainty in user load. The REIES system is affected by the uncertainty in both user load and renewable energy sources.

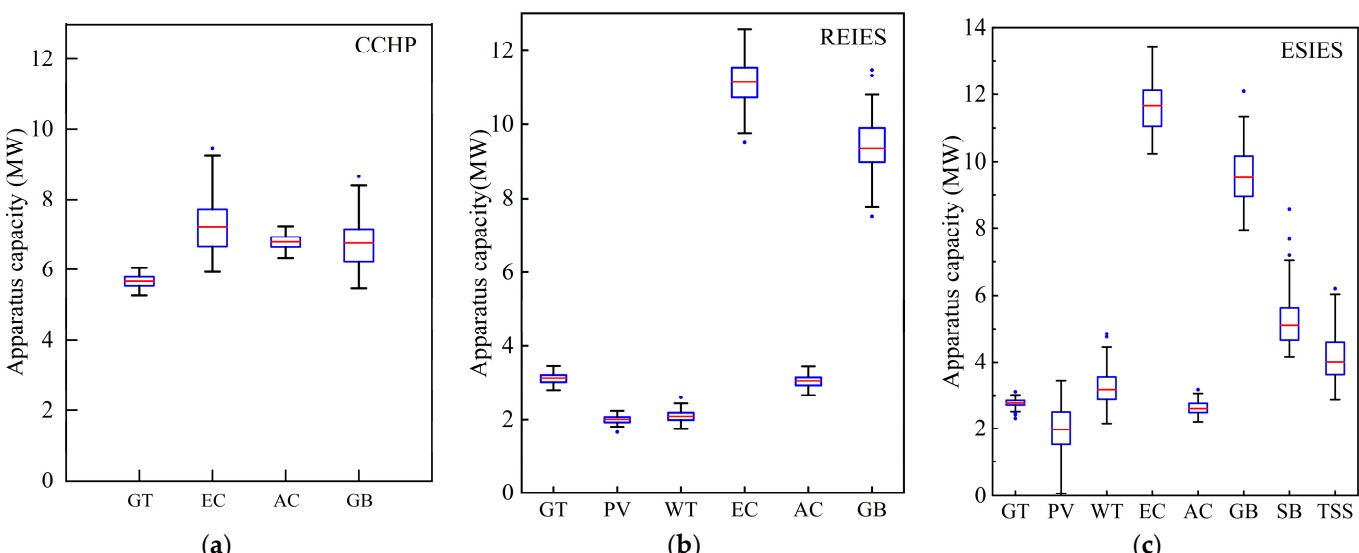

**Figure 15.** (**a**) Fluctuation of optimal equipment capacity for CCHP; (**b**) Fluctuation of optimal equipment capacity for REIES; (**c**) Fluctuation of optimal equipment capacity for ESIES.

It can be observed that the devices with the most significant fluctuations in optimal capacity are the electric chiller and gas boiler, while the capacities of other devices exhibit more minor fluctuations. Even devices like wind turbines and PV cells, which are directly impacted by uncertainty, show significantly lower capacity fluctuations than the electric

chiller and gas boiler. This indicates that the gas turbine and renewable electricity generation systems are used to meet stable customer demand, while the electric chiller and absorption chiller serve as supplements.

Therefore, in the CCHP design, increasing the capacity of the electric chiller by 5.8% and the absorption chiller by 6.2% can partially mitigate the negative impact of source-load uncertainty. For the REIES system, these values are 4.0% and 5.8%, respectively.

The fluctuation amplitude in the CCHP system is more significant than that in the REIES system, indicating that integrating renewable energy sources can reduce the impact of uncertainty on optimal equipment capacities. In the ESIES system, the optimal capacity fluctuations for each device are more extensive than in the previous two systems. This suggests that source-load uncertainty has introduced higher levels of uncertainty into the system, increasing the complexity of energy system design.

Penetration Rate of Renewable Energy

For supply security reasons, integrated energy systems still need conventional fossil energy sources. Therefore, in this section, the penetration rate of renewable energy (RER) is varied based on the previous optimization model to study its impact on the integrated energy system.

When the RER is zero, the energy system is essentially a CCHP system, and it is only influenced by fluctuations in user demand. Figure 17 reflects the situation in the CCHP system where both the gas turbine and the grid supply electricity. The power supply from the gas turbine shows relatively small fluctuations on four typical days. On the typical summer day, the proportion of electricity supplied by the grid is higher than on other typical days, and the fluctuations are more pronounced. This indicates that, when both the gas turbine and the grid supply electricity to users, the gas turbine handles the primary load, while the grid provides auxiliary power and deals with the uncertainty brought by fluctuations in user demand, reducing the fluctuations in gas turbine power supply. This is because the gas turbine operates in a combined heat and power mode. When there is both thermal and electrical demand in the system, the gas turbine takes priority in the energy supply system.

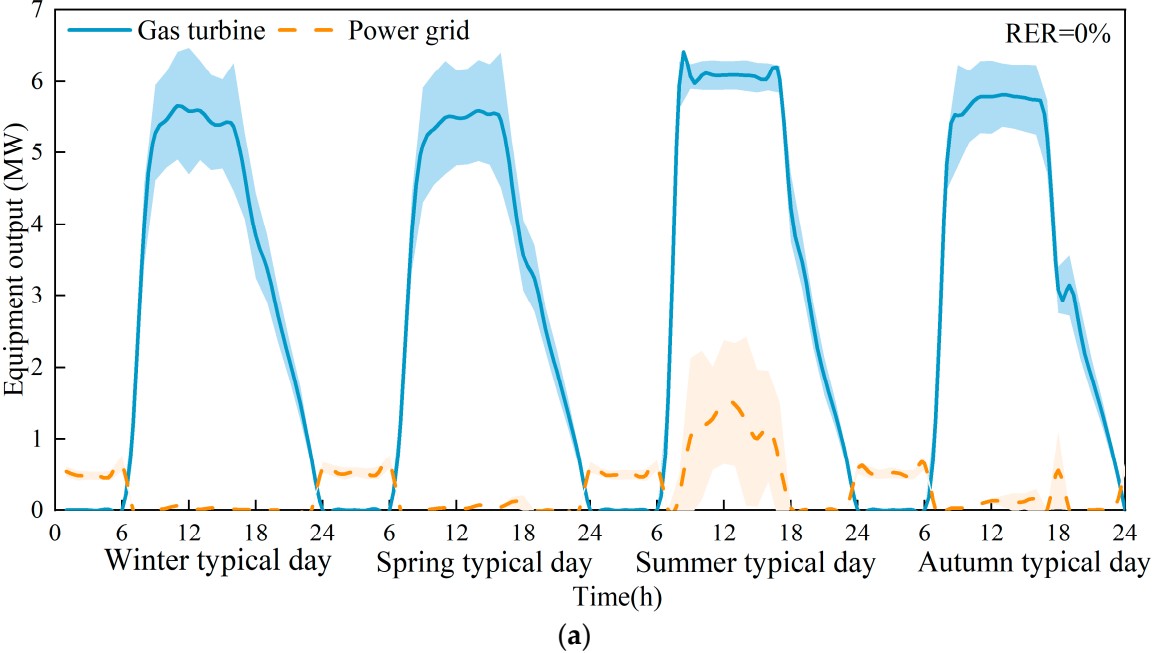

**(a)**

**Figure 16.** *Cont.*

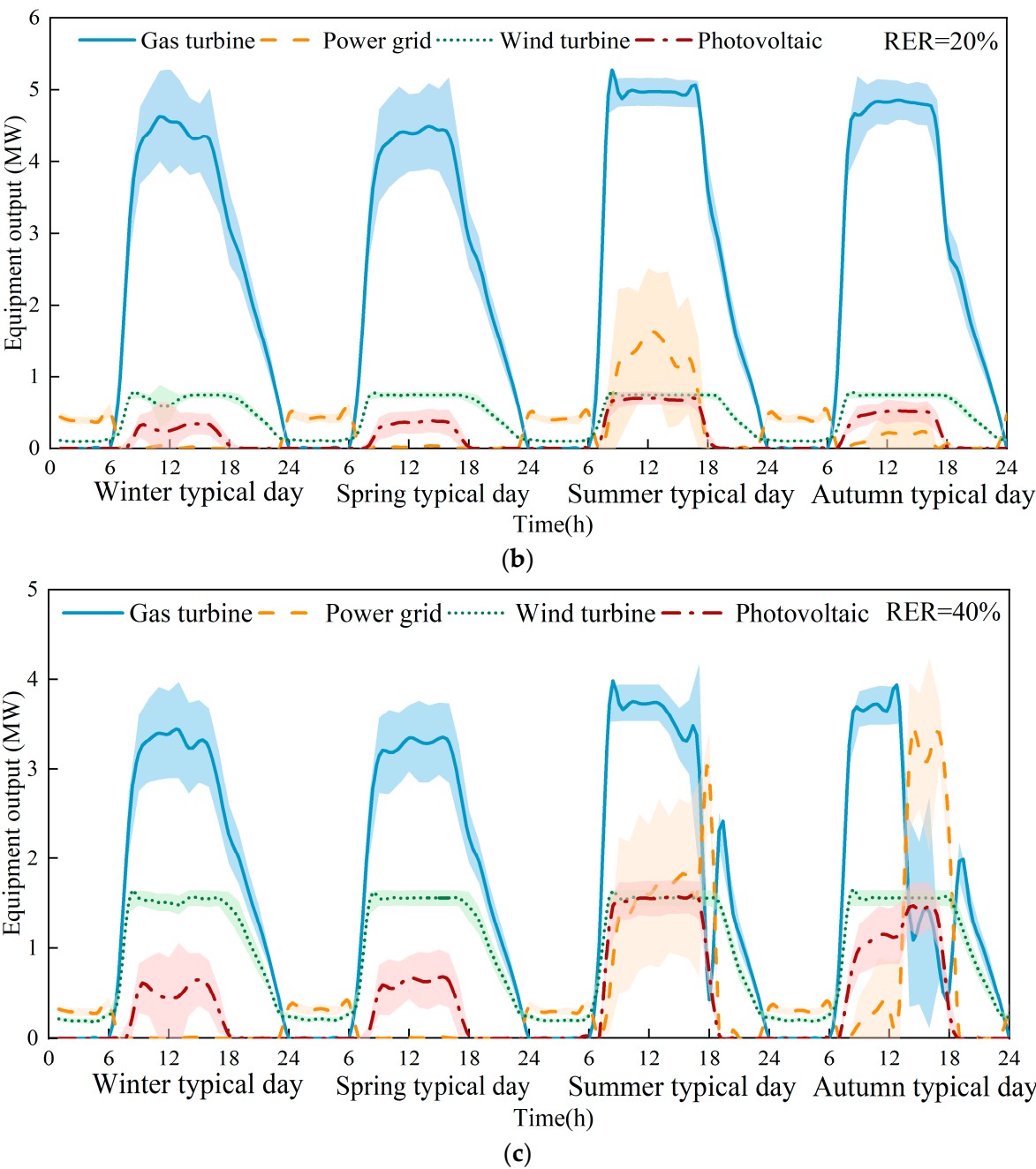

**Figure 17.** *Cont.*

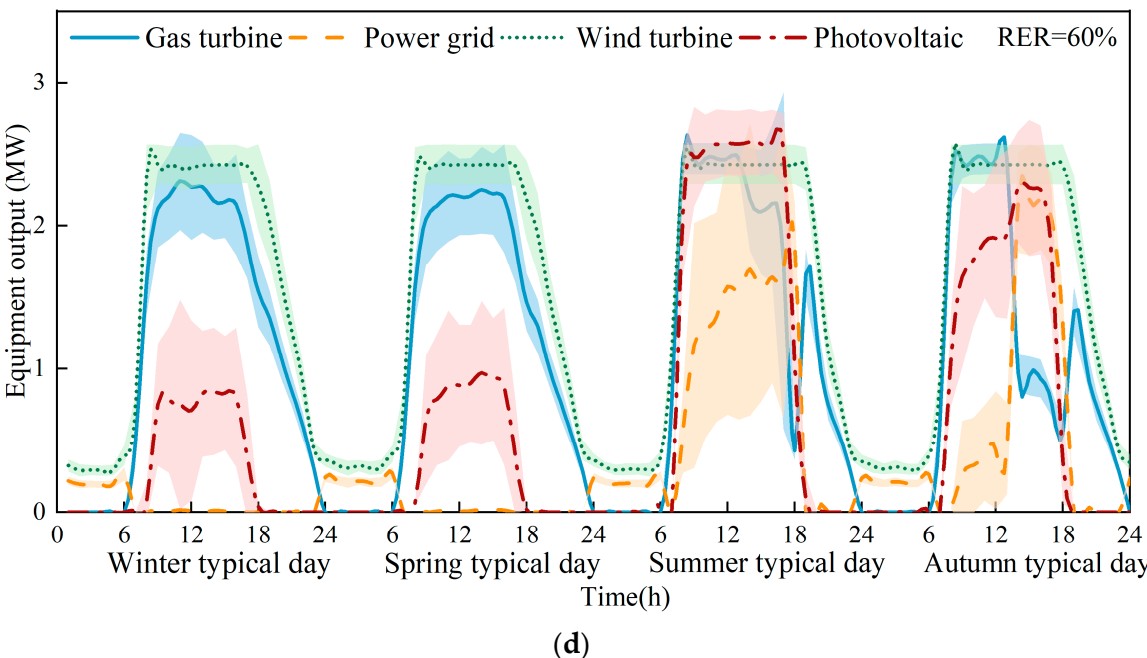

**Figure 17.** REIES equipment output uncertainty. (**a**) The RER is 0%. (**b**) The RER is 20%. (**c**) The RER is 40%. (**d**) The RER is 60%.

In the REIES system, the power output fluctuations of the gas turbine and the grid exhibit similar characteristics to those in the CCHP system. As the RER increases, the output of the gas turbine decreases, while the output of wind and solar power increases. The primary reason for the expanded fluctuation range in wind and solar power generation is the uncertainty associated with renewable energy sources.

The gas turbine's output fluctuation does not increase with the higher RER. Instead, it shows a decreasing trend due to the reduced output of the gas turbine. The fluctuation amplitude of grid electricity remains relatively stable. This indicates that while the grid may help the gas turbine handle some fluctuations, the increased RER has not changed the system's dependence on the grid or increased the fluctuation amplitude of its output.

## 5. Conclusions

This study focuses on an industrial park in Xi'an, China, and aims to develop a linear programming model with ATC as the objective function. Economic, energy efficiency, and environmental evaluation criteria are established to assess system performance. Sensitivity analysis is conducted to investigate the impact of critical parameters on the operation of the integrated energy system. Additionally, considering the integration of wind and solar energy, Monte Carlo simulations are employed to study the effects of uncertainty on system operation. The major conclusions can be summarized as follows:

- Based on the simulation results, it is found that the promotion or constraint relationship between renewable energy supply and energy cascade utilization depends on the relative sizes of the user's load thermal–electric ratio and the prime mover's thermal–electric ratio. Specifically, when the user's load thermal–electric ratio is greater than the prime mover's thermal–electric ratio, renewable energy and energy storage devices reduce the capacity of the cogeneration unit, leading to a constrained relationship between renewable energy supply and energy cascade utilization. Conversely, it exhibits a promotion relationship.

- The two metrics, carbon dioxide emissions, and system efficiency, are sensitive to natural gas and electricity prices. Therefore, reasonably setting natural gas and electricity prices can help improve the benefits of the system. When the electricity price exceeds 2.63 times the gas price, the increase in electricity price has almost no significant impact

on the system's cost. Similarly, when the gas price surpasses 0.63 times the electricity price, the rise in gas price contributes only marginally to the system's cost escalation. Therefore, in the design of energy systems, careful attention should be paid to the relative levels of electricity and natural gas prices to avoid negative impacts on the system due to energy price fluctuations and to enhance system performance.

- Equipment capacity is not sensitive to electricity, gas, and electricity buyback price fluctuations. Therefore, when designing system capacity, there is no need to pay too much attention to changes in energy prices. Regardless of price fluctuations in electricity and natural gas, the equipment capacity for gas boilers and electric chillers stays at a minimum of approximately 6MW. Consequently, when energy prices fluctuate, the impact on operational strategy design is more significant than capacity design.
- The uncertainty of renewable energy poses more significant challenges for the design of REIES systems. To cope with the negative impact of source-load uncertainty on the stable operation of the IES, the capacities of the electric chiller and absorption chiller should be increased by 4.0% and 5.8%, respectively. It is worth noting that the increase in the RER has not changed the system's dependence on the grid.

This study aims to provide fundamental research results for the operation of IESs coupled with renewable energy. Focusing on ATC as the optimization objective and establishing environmental and energy efficiency assessment criteria, this study facilitates the cost-effective operation of IESs. It aligns with sustainable development policies and has the potential to incentivize environmentally responsible businesses. Despite its contributions, there are still certain limitations due to the constraints and numerous decision variables in the model, which require using linear models for computational efficiency. However, in practical operation, uncertainties can arise from equipment operating under non-design conditions, and renewable energy generation is a significant source of instability in IES. The Monte Carlo method employed in this paper has some disparities with data collected from real-world processes, which is the primary factor affecting the model's robustness. Therefore, in future research efforts: (1) robustness of the model should be considered, and the study of interrelations among uncertainties can help address their adverse effects on the system. (2) Discussing the applications of IES coupled with renewable energy in industrial and residential sectors is beneficial and will create a sustainable future.

**Author Contributions:** Conceptualization, X.L. and W.Z.; methodology, X.L. and W.Z.; software, S.Y. and Y.J.; validation, Y.J., Z.G. and X.L.; formal analysis, S.Y., Y.J. and Z.G.; investigation, Y.J. and X.L.; resources, Y.J.; data curation, S.Y., Y.J. and Y.C.; writing—original draft preparation, X.L.; writing— review and editing, W.Z.; visualization, X.L. and Y.J.; supervision, W.Z.; project administration, W.Z.; funding acquisition, W.Z. All authors have read and agreed to the published version of the manuscript.

**Funding:** This study was supported by the National Key Research and Development Program of China [grant number 2017YFA0700300].

**Institutional Review Board Statement:** Not applicable.

**Informed Consent Statement:** Not applicable.

**Data Availability Statement:** Not applicable.

**Conflicts of Interest:** The authors declare no conflict of interest.

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
