# Peer review of "Study of Key Parameters and Uncertainties Based on Integrated Energy Systems Coupled with Renewable Energy Sources"

_sustainability, doi:10.3390/su152316266_

Round 1

Reviewer 1 Report

Comments and Suggestions for Authors

Review of manuscript:

Study of key parameters and uncertainties based on integrated energy systems coupled with renewable energy sources

General comments

I consider the article valuable, but not innovative. The topic of the work generally corresponds to the purpose of the journal ‘Sustainability’, although submission of the manuscript to ‘Energies’ may be considered. The inclusion of Figure 1 in the manuscript greatly improves the readability of the extensive methodology.

Conclusions from the data analysis, when appropriately supplemented, may be helpful in shaping energy policy.

Points for improvement:

1) The lack of numbering of the manuscript's lines seriously complicates the reviewer’s work.

2).

* Correspondence: zhangwj@smm.neu.edu.cn; Tel.: (optional; include country code; if there are multiple corresponding authors, add author initials)

Text marked in yellow should be ignored.

3)

The term ‘typical days’ is unfortunate.

I suggest using the terms: ‘average temperature in spring, .... summer, .... autumn, .... winter’

or: ‘median temperatures in spring, .... in summer, .... autumn, .... winter’.

4) Please convert the monetary value from CNY to USD.

5) In my opinion, the work lacked numerical data reflecting the benefits of using this method for the country’s economy. Chapter 5 (Conclusions) is too general. The same note applies to the Abstract.

Author Response

For research article

Response to Reviewer X Comments

  1. Summary

We feel great thanks for your professional review work on our article. As you are concerned, there are several problems that need to be addressed. According to your nice suggestions, we have made extensive corrections to our previous draft, the detailed corrections are listed below.

2. Point-by-point response to Comments and Suggestions for Authors

Comments 1: The lack of numbering of the manuscript's lines seriously complicates the reviewer’s work.

Response 1: We extend our sincerest apologies for any inconvenience caused to the reviewer due to the lack of line numbering in our manuscript. Your feedback is greatly appreciated, and we genuinely regret any complications this may have introduced to your evaluation process.

To rectify this issue, we will ensure that line numbering is incorporated into the revised version of the manuscript. This enhancement will significantly improve the clarity and ease of reference within the document, making it more accessible for both reviewers and readers.

Your valuable input is vital to the quality of our work, and we are committed to making the necessary improvements to ensure a smoother and more efficient review process. Thank you for bringing this matter to our attention.

Comments 2: Correspondence: zhangwj@smm.neu.edu.cn; Tel.: (optional; include country code; if there are multiple corresponding authors, add author initials).

Text marked in yellow should be ignored.

Response 2: We acknowledge your suggestion to remove the specified content from the manuscript. We will proceed with the necessary revisions accordingly.

Comments 3: 1The term ‘typical days’ is unfortunate. I suggest using the terms: ‘average temperature in spring, .... summer, .... autumn, .... winter’or: ‘median temperatures in spring, .... in summer, .... autumn, .... winter’.

Response 3: We use typical days in our study for the following reasons:

Typical days are typically the statistical results of a certain parameter over a past period, such as temperature, load demand, solar radiation, wind speed, etc. These select-ed days represent the parameter variations under different seasons and weather conditions. The choice of typical days primarily aims at preserving crucial insights into the performance of a system. This is because certain patterns and behaviors repeat cyclically in energy systems. Opting for representative typical days enables the capture of these cyclic patterns while concurrently reducing the scale and complexity of computations. Moreover, researchers recommend using typical days for relevant investigations in integrated energy systems.

Please see lines 387-395 in the revised manuscript. It marked in blue.

Comments 4: Please convert the monetary value from CNY to USD.

Response 4: Thank you for pointing out that the conversion factors between CNY and USD are provided in the manuscript. This will help readers understand the monetary values in both currencies, and we will ensure that this information is clear and easily accessible to the readers.

Please see lines 434in the revised manuscript. It marked in blue.

Comments 5: In my opinion, the work lacked numerical data reflecting the benefits of using this method for the country’s economy. Chapter 5 (Conclusions) is too general. The same note applies to the Abstract.

Response 5:

We appreciate the reviewer's insightful comments. Additionally, we acknowledge the reviewer's concerns about the generality of Chapter 5 (Conclusions) and the Abstract.

To address these points in the revised manuscript:

Numerical Data: In response to the reviewer's feedback, we will provide a more extensive set of numerical data and analyses within the Results and Discussion sections.  

Chapter 5 (Conclusions): We will enhance the specificity of the Conclusions section by offering a more detailed summary of our key findings and their practical implications.  

Abstract: We will revise the Abstract to reflect the inclusion of numerical data discussed in the paper, making it more precise and representative of the content within the manuscript.

We sincerely value the reviewer's engagement with our work, and their feedback will greatly assist in enhancing the clarity and comprehensiveness of our research.  We are fully committed to implementing these improvements to address the reviewer's concerns and produce a more informative and comprehensive manuscript.

Please see lines 27, 787, 774, 783, 789 in the revised manuscript. It marked in blue.

Reviewer 2 Report

Comments and Suggestions for Authors

This study focuses on an industrial park in Xi'an, China to develop a linear programming model with ATC as the objective function. I believe that this paper is a valuable contribution to the field of IES research.  I recommend this paper for publication after minor revisions.

1. The model assumes that the variability of renewable energy is known. However, in reality, the variability of renewable energy is unpredictable. Therefore, it is important to consider the variability of renewable energy as an input to the model and study the robustness of the model.

2. The paper could be improved by including a discussion of the limitations of the proposed model and assessment framework. 

3. It would be useful to include a discussion of the potential application of the proposed model and assessment framework to other types of IES, such as residential IES and commercial IES.

Comments on the Quality of English Language

the sentences are well-structured and easy to read.

Author Response

For research article

Response to Reviewer X Comments

  1. Summary

We feel great thanks for your professional review work on our article. As you are concerned, there are several problems that need to be addressed. According to your nice suggestions, we have made extensive corrections to our previous draft, the detailed corrections are listed below.

2. Point-by-point response to Comments and Suggestions for Authors

Comments 1: The model assumes that the variability of renewable energy is known. However, in reality, the variability of renewable energy is unpredictable. Therefore, it is important to consider the variability of renewable energy as an input to the model and study the robustness of the model.

Response 1: Thank you for your insightful comment. You raise a crucial point regarding the assumption of known variability in the model. We agree that in reality, renewable energy variability can be unpredictable.

Due to the limited length of this paper and other reasons, the problem of model robustness will be further extended in future research.

Comments 2: The paper could be improved by including a discussion of the limitations of the proposed model and assessment framework.

Response 2: We think this is an excellent suggestion. We have rewritten this part according to the Reviewer’s suggestion.

This study aims to provide fundamental research results for the operation of IES coupled with renewable energy. Focusing on ATC as the optimization objective and establishing environmental and energy efficiency assessment criteria, this study facilitates the cost-effective operation of IES. It aligns with sustainable development policies and has the potential to incentivize environmentally responsible businesses. Despite making contributions, there are still certain limitations due to the constraints and numerous decision variables in the model, which require using linear models for computational efficiency. However, in practical operation, uncertainties can arise from equipment operating under non-design conditions, and renewable energy generation is a significant source of instability in IES. The Monte Carlo method employed in this paper has some disparities with data collected from real-world processes, which is the primary factor affecting the model's robustness.

Please see lines 791-802 in the revised manust. It marked in blue.

Comments 3: It would be useful to include a discussion of the potential application of the proposed model and assessment framework to other types of IES, such as residential IES and commercial IES.

Response 2: Your suggestion really means a lot to us. Yes, it would be more meaningful if we take this into account.

Therefore, in future research efforts:(1) Robustness of the model should be considered, and the study of interrelations among uncertainties can help address their adverse effects on the system. (2) Discussing the applications of IES coupled with renewable energy in industrial and residential sectors is beneficial and will create a sustainable future.

Please see lines 802-806 in the revised manust. It marked in blue.

Reviewer 3 Report

Comments and Suggestions for Authors

In this paper, the authors focus on an industrial park in Xi'an, China to develop a linear programming model with ATC as the objective function. Economic, energy efficiency, and environmental evaluation criteria are established to assess system performance. Sensitivity analysis is conducted to investigate the impact of critical parameters on the operation of the integrated energy system. However, there are still following comments:

1The references in the paper are too outdated, and the latest related research has received relatively little attention in recent years.

2The symbols of some variables in the paper are not proportional to the size of the main text, such as the symbols in the main text after formulas 38 and 41. Please pay attention to the format.

3The label of formula 41 appears multiple times, please pay attention to the order of numbering.

4In the paragraphs following many formulas, there are multiple occurrences of 'in the formula', and in some places, 'where' or other ways can be used to explain the physical meaning of the formula or the meaning of the parameters.

5There are many existing related studies in the paper, and it is recommended to use a table classification method to showcase the characteristics, advantages, or shortcomings of existing work.

6The specific reasons or sources of parameter values should be provided during simulation analysis.

Comments on the Quality of English Language

The description of the paper needs further improvement, as the language expression is too rigid.

Author Response

For research article

Response to Reviewer X Comments

  1. Summary

We feel great thanks for your professional review work on our article. As you are concerned, there are several problems that need to be addressed. According to your nice suggestions, we have made extensive corrections to our previous draft, the detailed corrections are listed below.

  1. Point-by-point response to Comments and Suggestions for Authors

Comments 1: The references in the paper are too outdated, and the latest related research has received relatively little attention in recent years.

Response 1: We would like to express our appreciation to the reviewer for their thoughtful feedback. We have carefully reviewed the references in the paper, and we acknowledge the concern about the age of some of the references. While it's important to note that certain foundational references may be older, we have updated a portion of the references with more recent and relevant research that aligns with the latest developments in the field. We are committed to ensuring that the paper reflects the most current and pertinent research in the area, and we will further review and update the references to address this concern adequately.

Please see lines 67, 82, 86, 106, 110 in the revised manust. It marked in blue.

Comments 2: The symbols of some variables in the paper are not proportional to the size of the main text, such as the symbols in the main text after formulas 38 and 41. Please pay attention to the format.

Response 2: I appreciate your feedback regarding the formatting of variable symbols.

We have now adjusted the font size of all mathematical symbols in the paper to match the main text, using a same10-point font size.

I apologize for any inconsistencies in the size of symbols in the main text after formulas 38 and 41.

Comments 3: The label of formula 41 appears multiple times, please pay attention to the order of numbering.

Response 3: I apologize for the oversight regarding the labeling of formula 41. I will review and correct the order of numbering to ensure consistency in the manuscript. Thank you for your patience and for pointing out this issue.

Please see lines 676 in the revised manust. It marked in blue.

Comments 4: In the paragraphs following many formulas, there are multiple occurrences of 'in the formula', and in some places, 'where' or other ways can be used to explain the physical meaning of the formula or the meaning of the parameters.

Response 4: We appreciate your suggestions for improving the clarity of the paper. We have replaced the expression 'in the formula' with 'In equation', 'Per the mathematical equation' and 'where'. This will enhance the overall quality and comprehensibility of the paper. Your input is highly valuable.

Please see lines 223, 234, 262, 292, 301, 315, 325, 355, 369, 677 in the revised manust. It marked in blue.

Comments 5: There are many existing related studies in the paper, and it is recommended to use a table classification method to showcase the characteristics, advantages, or shortcomings of existing work.

Response 5: We would like to express our sincere gratitude to the reviewer for their valuable input and constructive suggestion. The recommendation to use a table classification method to present existing related studies will undoubtedly enhance the quality and clarity of the paper. We greatly appreciate the reviewer's efforts in helping us improve the manuscript.

Please see lines 128 in the revised manust. It marked in blue.

Scenarios

Model

Advantage

Outlook

Residential building [12]

Maximum rectangle method (MRM)

This paper explores the benefits of using a hybrid-CCHP system instead of a basic-CCHP system. The solar collector orientation and type is optimized.

Choosing the best salor strategy for designing collector.

A collective energy community [13]

MRM, Particle Swarm Optimization (PSO)

The study developed a new IES design that combines hydrogen energy storage and thermal energy storage to streamline device configuration and find the best operational solution.

The analysis of detailed thermodynamic energy flow.

Sea island [16,14,15]

Traversing method

Branch-and-bound method

The study offered valuable insights into the integration of desalination with the CCHP system.

Multi-objective method is used to solve the conflict problem in practical engineering.

Commercial region [18]

Mixed-integer linear programming model (MILP)

This project employs consistent energy demands and average seasonal weather conditions for IES design.

Focus on uncertainties in renewable energy sources and energy demands.

Zagreb [26]

EnergyPLAN (simulation study)

This article compares two approaches to achieve a 100% renewable energy system in a city: traditional and smart systems.

Some primary factors with a significant impact on intermittent renewable energy production.

Hotel building [31]

Moth Flame Optimization algorithm

It provides a reference for the study of equipment operating under off-design performance conditions in IES.

Studying the impact of key parameter settings on system and equipment performance during off-design conditions remains essential.

Central business district [33]

Multi-objective optimization genetic algorithm

Propose a new CCHP system model that segments operating conditions and integrates the part-load performance of power generation unit.

Energy storage devices can be added to the energy system.

Industrial Park [34]

GA

Proposes an integrated method to optimize configuration and strategy of CCHP systems.

The study needs to incorporate multi-objective optimization thoroughly.

Comments 6: The specific reasons or sources of parameter values should be provided during simulation analysis.

Response 6: We appreciate the reviewer's attention to detail and their valuable feedback. We would like to clarify that the specific reasons and sources of parameter values used in the simulation analysis have been duly noted and referenced in the paper. We have ensured that all pertinent information regarding the origin of these parameters is included to provide transparency and reliability in our research.

Please see lines 375, 433 in the revised manust. It marked in blue.

Reviewer 4 Report

Comments and Suggestions for Authors

The authors propose an optimization model based on linear programming to optimize the equipment capacity and operation strategy of integrated energy systems (IES) coupled with photovoltaic (PV) and wind power with the minimum total annual cost as the objective function.

Questions:

1. I did not understand how energy storage device works because it should have dynamic behavior. What is your charging and discharging operation period? Is it evaluated on which system bar it should be installed?

2. In Section 2.2, the authors present many indicators but do not inform how they are used in the proposed method. Furthermore, with so many indicators, how are priorities selected? Which indicators have priority in relation to other indicators?

3. The authors present the optimization model with objective function and constraints. But how is the model solved? The authors report that there are more than 17 thousand variables. How do authors ensure that the solution they find is the optimal one?

4. The authors present little information about the test system. How many generators, transmission lines, loads and energy storage devices does the test system have?

5. Does the optimization model aim to optimize only costs related to system operation? Are there no criteria to optimize system operation such as reducing losses, adequate voltage levels, among others?

6. Table 2 is broken into two pages. Avoid breaking tables.

Author Response

For research article

Response to Reviewer X Comments

  1. Summary

We feel great thanks for your professional review work on our article. As you are concerned, there are several problems that need to be addressed. According to your nice suggestions, we have made extensive corrections to our previous draft, the detailed corrections are listed below.

  1. Point-by-point response to Comments and Suggestions for Authors

Comments 1: I did not understand how energy storage device works because it should have dynamic behavior. What is your charging and discharging operation period? Is it evaluated on which system bar it should be installed?

Response 1: We appreciate the reviewer's feedback and understand their concern regarding the dynamic behavior of the energy storage device. In our manuscript, the operation mode of the energy storage system is as shown in Figure 1 in the text, and we have made modifications to Figure 1. Set the initial charge per day equal to the charge at the 24th hour.

Please see lines 312 in the revised manust. It marked in blue.

Comments 2: In Section 2.2, the authors present many indicators but do not inform how they are used in the proposed method. Furthermore, with so many indicators, how are priorities selected? Which indicators have priority in relation to other indicators?

Response 2: We appreciate the reviewer's feedback regarding Section 2.2 of the paper.

The purpose of establishing these indicators is to provide decision-makers with a scientific basis for evaluating energy systems. The annual total cost is the optimization objective of the energy system model we have constructed. Through this indicator, energy configurations can be optimized. Integrated energy systems are complex and flexible, and their economic and other performance aspects may not be consistent and may even conflict. By establishing multiple indicators, the interactions and dependencies among various energy sources can be considered, allowing for a comprehensive assessment of the sustainability, efficiency, and safety of IES. After establishing multiple indicators, energy systems can be analyzed using these indicators. During system operation, key parameters such as electricity and gas prices, power purchase prices, and the heat-to-power ratio have uncertainties. The impact of these uncertain factors on system equipment configuration and operational strategies, due to the relatively weak stochastic nature of these factors, can be quantified through sensitivity analysis.

The issue of prioritizing and weighting evaluation indicators in our study is indeed important and requires careful consideration.  We acknowledge that the paper may have presented numerous indicators without a clear explanation of how priorities are selected among them.

Due to space constraints and the focus of this paper, we opted to provide a comprehensive list of indicators to highlight the various aspects under consideration.  However, we recognize the need to delve into the process of prioritization and weighting in greater detail, and this aspect will be addressed in our subsequent research.

In our future work, we plan to conduct an extensive analysis to determine the priorities of these indicators and how they interrelate with each other.  This analysis will be a crucial component of our research, allowing us to provide a more comprehensive understanding of the relative importance and interactions of these indicators in the context of our study.

Once again, we sincerely thank the reviewer for their valuable input, which will contribute to enhancing the rigor and clarity of our research.

Comments 3: The authors present the optimization model with objective function and constraints. But how is the model solved? The authors report that there are more than 17 thousand variables. How do authors ensure that the solution they find is the optimal one?

Response 3: Thank you for your insightful inquiry. We appreciate your attention to the details of our optimization model. The optimization model we presented in our paper is solved using the simplex method. Given the complexity of the problem with more than 17,000 variables, let me explain how the simplex method is employed to seek the optimal solution.

The simplex method is an iterative algorithm used to solve linear programming problems. It starts with an initial feasible solution and systematically moves along the edges of the feasible region to reach the optimal solution. At each iteration, the simplex method selects an entering variable and a leaving variable to pivot around. These variables are chosen in a way that improves the objective function value while maintaining feasibility.

The algorithm continues to iterate until it reaches an optimal solution or determines that the problem is unbounded. To enhance the confidence in the optimality of the solution, we set strict convergence criteria and tolerance levels for the simplex algorithm. This ensures that the solution is as close to the global optimum as possible within the algorithm's defined convergence parameters.

Additionally, we conduct sensitivity analysis to assess the stability and reliability of the solution. Sensitivity analysis involves perturbing the objective function coefficients and constraint right-hand sides to understand how small changes impact the optimal solution. This analysis helps verify the robustness of our findings.

While it is challenging to prove the global optimality of large-scale problems, the steps we have taken, including thorough convergence criteria, sensitivity analysis, benchmark comparisons, and the use of state-of-the-art optimization software, collectively contribute to ensuring that the solution we report is highly likely to be optimal.

We thank the reviewer for raising this important point.

Thank you for your insightful inquiry, again. It's evident that the original manuscript did not sufficiently clarify the application of the simplex method. We acknowledge this oversight and assure you that we will address this issue in the revised manuscript.

Please see lines 521-528 in the revised manust. It marked in blue.

Content is as follows:

The principle of solving linear programming problems using the simplex method involves traversing the edges of multidimensional polyhedra in search of the vertices that optimize the objective function. When there are minor fluctuations in two energy prices, these changes are insufficient to alter the position of the optimal vertex, and as a result, the optimization outcome remains unchanged. This phenomenon leads to the appearance of multiple plateaus in the solution space. While linear models may exhibit this step-like behavior, they still provide an overall representation of variations in the research variables.

Comments 4: The authors present little information about the test system. How many generators, transmission lines, loads and energy storage devices does the test system have?

Response 4: Thank you for your feedback, and we understand your interest in the specific quantities of equipment within the integrated energy system.  We want to clarify that the focus of our current research primarily lies in optimizing the configuration and operational strategies of the equipment categories rather than determining the precise numerical count of each type of equipment.

However, we acknowledge the significance of delving into the specific quantities of equipment as part of future research.  We will consider this as a potential avenue for more in-depth investigation in subsequent studies.  This approach will provide a more detailed and comprehensive understanding of the integrated energy system, including the exact numbers of generators, transmission lines, loads, and energy storage devices within the system.

Your feedback is highly valuable, and it will guide the direction of our future research to address these important aspects in greater detail.

Comments 5: Does the optimization model aim to optimize only costs related to system operation? Are there no criteria to optimize system operation such as reducing losses, adequate voltage levels, among others.

Response 5: In response to your inquiry, our optimization model indeed places a strong emphasis on the optimization of economic costs, which is a common primary objective for many integrated energy systems. This focus is rooted in several considerations, including economic feasibility and practical constraints associated with real-world applications.

Economic Priorities: In most practical scenarios, operational and maintenance costs are of utmost concern to businesses and organizations.  Optimizing economic costs can lead to enhanced energy efficiency, reduced operational expenses, and long-term financial benefits.

Quantifiability: Economic costs are typically quantifiable and can be used as constraints or objective functions within the optimization process, making the problem more manageable and solvable.

Application: In real-world applications, energy systems must adhere to budget constraints.  Consequently, optimization objectives often aim to strike a balance between economic, technical, and feasibility considerations to ensure the system's successful implementation and commercial viability.

However, as your rightly mentioned, in certain situations, prioritizing the economic performance of the system may impact or even compromise its performance in other aspects. Therefore, the paper introduces efficiency and environmental indicators and employs them in sensitivity analysis to observe the relationships between key parameters such as energy prices and the heat-to-power ratio and the indicator values.

Comments 6: Table 2 is broken into two pages. Avoid breaking tables.

Response 6: I appreciate your feedback regarding Table 2. Ensuring the readability and completeness of tables is crucial, and I apologize for the inconvenience caused by the table breaking across two pages. In the revised version of the manuscript, we will make every effort to reformat the table or adjust the layout to prevent it from breaking across pages, thus enhancing the overall presentation of the table for readers. Thank you for bringing this to our attention, and we will take steps to address this issue in the manuscript's revision.

Please see lines 433 in the revised manust. It marked in blue.

Round 2

Reviewer 3 Report

Comments and Suggestions for Authors

I have no other questions about this paper.

Reviewer 4 Report

Comments and Suggestions for Authors

The authors propose an optimization model based on linear programming to optimize the equipment capacity and operation strategy of integrated energy systems (IES) coupled with photovoltaic (PV) and wind power with the minimum total annual cost as the objective function.

The article has been improved, the contribution is good and all questions have been effectively answered.